# Iterative Methods for Private Synthetic Data: Unifying Framework and New Methods

**Terrance Liu**[*]
Carnegie Mellon University
Pittsburgh, PA 15213
terrancl@andrew.cmu.edu

**Giuseppe Vietri**[*]
University of Minnesota
Minneapolis, MN 55455
vietr002@umn.edu

**Zhiwei Steven Wu**
Carnegie Mellon University
Pittsburgh, PA 15213
zstevenwu@cmu.edu

## Abstract

We study private synthetic data generation for query release, where the goal is to construct a sanitized version of a sensitive dataset, subject to differential privacy, that approximately preserves the answers to a large collection of statistical queries. We first present an algorithmic framework that unifies a long line of iterative algorithms in the literature. Under this framework, we propose two new methods. Our first method, generative networks with the exponential mechanism (GEM), circumvents computational bottlenecks in algorithms such as MWEM by optimizing over generative models parameterized by neural networks, which capture a rich family of distributions while enabling fast gradient-based optimization. The second method, private entropy projection (PEP), can be viewed as an advanced variant of MWEM that adaptively reuses past query measurements to boost accuracy. We demonstrate that GEM and PEP empirically outperform existing algorithms. Furthermore, we show that GEM nicely incorporates prior information from public data while overcoming limitations of PMW^Pub, the existing state-of-the-art method that also leverages public data.

## 1 Introduction

As the collection and analyses of sensitive data become more prevalent, there is an increasing need to protect individuals' private information. Differential privacy [15] is a rigorous and meaningful criterion for privacy preservation that enables quantifiable trade-offs between privacy and accuracy. In recent years, there has been a wave of practical deployments of differential privacy across organizations such as Google, Apple, and most notably, the U.S. Census Bureau [3].

In this paper, we study the problem of differentially private query release: given a large collection of statistical queries, the goal is to release approximate answers subject to the constraint of differential privacy. Query release has been one of the most fundamental and practically relevant problems in differential privacy. For example, the release of summary data from the 2020 U.S. Decennial Census can be framed as a query release problem. We focus on the approach of synthetic data generation—that is, generate a privacy-preserving "fake" dataset, or more generally a representation of a probability distribution, that approximates all statistical queries of interest. Compared to simple Gaussian or Laplace mechanisms that perturb the answers directly, synthetic data methods can provably answer an exponentially larger collection of queries with non-trivial accuracy. However, their statistical advantage also comes with a computational cost. Prior work has shown that achieving better accuracy than simple Gaussian perturbation is intractable in the worst case even for the simple query class of 2-way marginals that release the marginal distributions for all pairs of attributes [38].

---

[*]First two authors contributed equally.

35th Conference on Neural Information Processing Systems (NeurIPS 2021).

Despite its worst-case intractability, there has been a recent surge of work on practical algorithms for generating private synthetic data. Even though they differ substantially in details, these algorithms share the same iterative form that maintains and improves a probability distribution over the data domain: identifying a small collection of high-error queries each round and updating the distribution to reduce these errors. Inspired by this observation, we present a unifying algorithmic framework that captures these methods. Furthermore, we develop two new algorithms, GEM and PEP, and extend the former to the setting in which public data is available. We summarize our contributions below:

**Unifying algorithmic framework.** We provide a framework that captures existing iterative algorithms and their variations. At a high level, algorithms under this framework maintain a probability distribution over the data domain and improve it over rounds by optimizing a given loss function. We therefore argue that under this framework, the optimization procedures of each method can be reduced to what loss function is minimized and how its distributional family is parameterized. For example, we can recover existing methods by specifying choices of loss functions—we rederive MWEM [21] using an entropy-regularized linear loss, FEM [40] using a linear loss with a linear perturbation, and DualQuery [19] with a simple linear loss. Lastly, our framework lends itself naturally to a softmax variant of RAP [5], which we show outperforms RAP itself.[2]

**Generative networks with the exponential mechanism** (GEM). GEM is inspired by MWEM, which attains worst-case theoretical guarantees that are nearly information-theoretically optimal [11]. However, MWEM maintains a joint distribution over the data domain, resulting in a runtime that is exponential in the dimension of the data. GEM avoids this fundamental issue by optimizing the absolute loss over a set of generative models parameterized by neural networks. We empirically demonstrate that in the high-dimensional regime, GEM outperforms all competing methods.

**Private Entropy Projection** (PEP). The second algorithm we propose is PEP, which can be viewed as a more advanced version of MWEM with an adaptive and optimized learning rate. We show that PEP minimizes a regularized exponential loss function that can be efficiently optimized using an iterative procedure. Moreover, we show that PEP monotonically decreases the error over rounds and empirically find that it achieves higher accuracy and faster convergence than MWEM.

**Incorporating public data.** Finally, we consider extensions of our methods that incorporate prior information in publicly available datasets (e.g., previous data releases from the American Community Survey (ACS) prior to their differential privacy deployment). While Liu et al. [26] has established PMW$^{\mathsf{Pub}}$ as a state-of-the-art method for incorporating public data into private query release, we discuss how limitations of their algorithm prevent PMW$^{\mathsf{Pub}}$ from effectively using certain public datasets. We then demonstrate empirically that GEM circumvents such issues via simple pretraining, achieving max errors on 2018 ACS data for Pennsylvania (at $\varepsilon = 1$) **9.23x** lower than PMW$^{\mathsf{Pub}}$ when using 2018 ACS data for California as the public dataset.

## 1.1 Related work

Beginning with the seminal work of Blum et al. [9], a long line of theoretical work has studied private synthetic data for query release [33, 21, 22, 20]. While this body of work establishes optimal statistical rates for this problem, their proposed algorithms, including MWEM [22], typically have running time exponential in the dimension of the data. While the worst-case exponential running time is necessary (given known lower bounds [16, 36, 39]), a recent line of work on practical algorithms leverage optimization heuristics to tackle such computational bottlenecks [19, 40, 5]. In particular, DualQuery [19] and FEM [40] leverage integer program solvers to solve their NP-hard subroutines, and RAP [5] uses gradient-based methods to solve its projection step. In Section 3, we demonstrate how these algorithms can be viewed as special cases of our algorithmic framework. Our work also relates to a growing line of work that use public data for private data analyses [6, 4, 7]. For query release, our algorithm, GEM, improves upon the state-of-the-art method, PMW$^{\mathsf{Pub}}$ [26], which is more limited in the range of public datasets it can utilize. Finally, our method GEM is related to a line

---

[2]We note that Aydore et al. [5] have since updated the original version (`https://arxiv.org/pdf/2103.06641v1.pdf`) of their work to include a modified version of RAP that leverages SparseMax [27], similar to way in which the softmax function is applied in our proposed baseline, RAP$^{\mathsf{softmax}}$.

of work on differentially private GANs [8, 44, 30, 42]. However, these methods focus on generating synthetic data for simple downstream machine learning tasks rather than for query release.

Beyond synthetic data, a line of work on query release studies "data-independent" mechanisms (a term formulated in Edmonds et al. [17]) that perturb the query answers with noise drawn from a data-independent distribution (that may depend on the query class). This class of algorithms includes the matrix mechanism [25], the high-dimensional matrix mechanism (HDMM) [28], the projection mechanism [31], and more generally the class of factorization mechanisms [17]. In addition, McKenna et al. [29] provide an algorithm that can further reduce query error by learning a probabilistic graphical model based on the noisy query answers released by privacy mechanisms.

## 2 Preliminaries

Let $\mathcal{X}$ denote a finite $d$-dimensional data domain (e.g., $\mathcal{X} = \{0,1\}^d$). Lete $U$ be the uniform distribution over the domain $\mathcal{X}$. Throughout this work, we assume a private dataset $P$ that contains the data of $n$ individuals. For any $x \in \mathcal{X}$, we represent $P(x)$ as the normalized frequency of $x$ in dataset $P$ such that $\sum_{x \in \mathcal{X}} P(x) = 1$. One can think of a dataset $P$ either as a multi-set of items from $\mathcal{X}$ or as a distribution over $\mathcal{X}$.

We consider the problem of accurately answering an extensive collection of linear statistical queries (also known as counting queries) about a dataset. Given a finite set of queries $Q$, our goal is to find a synthetic dataset $D$ such that the maximum error over all queries in $Q$, defined as $\max_{q \in Q} |q(P) - q(D)|$, is as small as possible. For example, one may query a dataset by asking the following: how many people in a dataset have brown eyes? More formally, a statistical linear query $q_\phi$ is defined by a predicate function $\phi : \mathcal{X} \to \{0,1\}$, as $q_\phi(D) = \sum_{x \in \mathcal{X}} \phi(x)D(x)$ for any normalized dataset $D$. Below, we define an important, and general class of linear statistical queries called $k$-way marginals.

**Definition 1** ($k$-way marginal). *Let the data universe with $d$ categorical attributes be $\mathcal{X} = (\mathcal{X}_1 \times \ldots \times \mathcal{X}_d)$, where each $\mathcal{X}_i$ is the discrete domain of the $i$th attribute $A_i$. A $k$-way marginal query is defined by a subset $S \subseteq [d]$ of $k$ features (i.e., $|S| = k$) plus a target value $y \in \prod_{i \in S} \mathcal{X}_i$ for each feature in $S$. Then the marginal query $\phi_{S,y}(x)$ is given by:*

$$\phi_{S,y}(x) = \prod_{i \in S} \mathbb{1}\left(x_i = y_i\right)$$

*where $x_i \in \mathcal{X}_i$ means the $i$-th attribute of record $x \in \mathcal{X}$. Each marginal has a total of $\prod_{i=1}^{k} |\mathcal{X}_i|$ queries, and we define a workload as a set of marginal queries.*

We consider algorithms that input a dataset $P$ and produce randomized outputs that depend on the data. The output of a randomized mechanism $\mathcal{M} : \mathcal{X}^* \to \mathcal{R}$ is a privacy preserving computation if it satisfies differential privacy (DP) [15]. We say that two datasets are neighboring if they differ in at most the data of one individual.

**Definition 2** (Differential privacy [15]). *A randomized mechanism $\mathcal{M} : \mathcal{X}^n \to \mathcal{R}$ is $(\varepsilon, \delta)$-differentially privacy, if for all neighboring datasets $P, P'$ (i.e., differing on a single person), and all measurable subsets $S \subseteq \mathcal{R}$ we have:*

$$\Pr\left[\mathcal{M}(P) \in S\right] \le e^\varepsilon \Pr\left[\mathcal{M}(P') \in S\right] + \delta$$

Finally, a related notion of privacy is called concentrated differential privacy (zCDP) [14, 10], which enables cleaner composition analyses for privacy.

**Definition 3** (Concentrated DP, Dwork and Rothblum [14], Bun and Steinke [10]). *A randomized mechanism $\mathcal{M} : \mathcal{X}^n \to \mathcal{R}$ is $\frac{1}{2}\tilde{\varepsilon}^2$-CDP, if for all neighboring datasets $P, P'$ (i.e., differing on a single person), and for all $\alpha \in (1, \infty)$,*

$$\mathrm{R}_\alpha\left(\mathcal{M}(P) \parallel \mathcal{M}(P')\right) \le \frac{1}{2}\tilde{\varepsilon}^2 \alpha$$

*where $\mathrm{R}_\alpha\left(\mathcal{M}(P) \parallel \mathcal{M}(P')\right)$ is the Rényi divergence between the distributions $\mathcal{M}(P)$ and $\mathcal{M}(P')$.*

## 3 A Unifying Framework for Private Query Release

In this work, we consider the problem of finding a distribution in some family of distributions $\mathcal{D}$ that achieves low error on all queries. More formally, given a private dataset $P$ and a query set $Q$, we

solve an optimization problem of the form:

$$\min_{D \in \mathcal{D}} \max_{q \in Q} |q(P) - q(D)| \tag{1}$$

In Algorithm 1, we introduce **Adaptive Measurements**, which serves as a general framework for solving (1). At each round $t$, the framework uses a private selection mechanism to choose $k$ queries $\widetilde{Q}_t = \{\widetilde{q}_{t,1}, \dots, \widetilde{q}_{t,k}\}$ with higher error from the set $Q$. It then obtains noisy measurements for the queries, which we denote by $\widetilde{A}_t = \{\widetilde{a}_{t,1}, \dots, \widetilde{a}_{t,k}\}$, where $\widetilde{a}_{t,i} = \widetilde{q}_{t,i} + z_{t,i}$ and $z_{t,i}$ is random Laplace or Gaussian noise. Finally, it updates its approximating distribution $D_t$, subject to a loss function $\mathcal{L}$ that depends on $\widetilde{Q}_{1:t} = \bigcup_{i=1}^{t} \widetilde{Q}_i$ and $\widetilde{A}_{1:t} = \bigcup_{i=1}^{t} \widetilde{A}_i$. We note that $\mathcal{L}$ serves as a surrogate problem to (1) in the sense that the solution of $\min_{D \in \mathcal{D}} \mathcal{L}(D)$ is an approximate solution for (1). We list below the corresponding loss functions for various algorithms in the literature of differentially private synthetic data. (We defer the derivation of these loss functions to the appendix.)

**MWEM from Hardt et al. [22]**   MWEM solves an entropy regularized minimization problem:

$$\mathcal{L}^{\mathsf{MWEM}}(D, \widetilde{Q}_{1:t}, \widetilde{A}_{1:t}) = \sum_{i=1}^{t} \sum_{x \in \mathcal{X}} D(x)\widetilde{q}_i(x)\left(\widetilde{a}_i - \widetilde{q}_i(D_{i-1})\right) + \sum_{x \in \mathcal{X}} D(x) \log D(x)$$

We note that $\mathsf{PMW}^{\mathsf{Pub}}$ [26] optimizes the same problem but restricts $\mathcal{D}$ to distributions over the public data domain while initializing $D_0$ to be the public data distribution.

**DualQuery from Gaboardi et al. [19]**   At each round $t$, DualQuery samples $s$ queries ($\widetilde{Q}_t = \{\widetilde{q}_{t,1}, \dots \widetilde{q}_{t,s}\}$) from $\mathcal{Q}_t$ and outputs $D_t$ that minimizes the the following loss function:

$$\mathcal{L}^{\mathsf{DualQuery}}(D, \widetilde{Q}_t) = \sum_{i=1}^{s} \widetilde{q}_{t,i}(D)$$

**FEM from Vietri et al. [40]**   The algorithm FEM employs a follow the perturbed leader strategy, where on round $t$, FEM chooses the next distribution by solving:

$$\mathcal{L}^{\mathsf{FEM}}(D, \widetilde{Q}_{1:t}) = \sum_{i=1}^{t} \widetilde{q}_t(D) + \mathbb{E}_{x \sim D, \eta \sim \mathrm{Exp}(\sigma)^d}\left(\langle x, \eta \rangle\right)$$

**RAP$^{\mathsf{softmax}}$ adapted from Aydore et al. [5]**   We note that RAP follows the basic structure of Adaptive Measurements, where at iteration $t$, RAP solves the following optimization problem:

$$\mathcal{L}^{\mathsf{RAP}}(D, \widetilde{Q}_{1:t}, \widetilde{A}_{1:t}) = \sum_{i,j} \left(\widetilde{q}_{i,j}(D) - \widetilde{a}_{i,j}\right)^2$$

However, rather than outputting a dataset that can be expressed as some distribution over $\mathcal{X}$, RAP projects the noisy measurements onto a continuous relaxation of the binarized feature space of $\mathcal{X}$, outputting $D \in [-1, 1]^{n' \times d}$ (where $n'$ is an additional parameter). Therefore to adapt RAP to Adaptive Measurements, we propose a new baseline algorithm that applies the softmax function instead of clipping each dimension of $D$ to be between $-1$ and $1$. For more details, refer to Section 4 and Appendix B, where describe how softmax is applied in GEM in the same way. With this slight modification, this algorithm, which we denote as RAP$^{\mathsf{softmax}}$, fits nicely into the

Adaptive Measurements framework in which we output a synthetic dataset drawn from some probabilistic family of distributions $\mathcal{D} = \left\{\sigma(M)|M \in \mathbb{R}^{n' \times d}\right\}$.

---

**Algorithm 1:** Adaptive Measurements

---

**Input:** Private dataset $P$ with $n$ records, set of linear queries $Q$, distributional family $\mathcal{D}$, loss functions $\mathcal{L}$, number of iterations $T$

Initialize distribution $D_0 \in \mathcal{D}$

**for** $t = 1, \ldots, T$ **do**

    **Sample**: For $i \in [k]$, choose $\widetilde{q}_{t,i}$ using a differentially private selection mechanism.

    **Measure:** For $i \in [k]$, let $\widetilde{a}_{t,i} = \widetilde{q}_{t,i}(P) + z_{t,i}$ where $z$ is Gaussian or Laplace noise

    **Update:** Let $\widetilde{Q}_t = \{\widetilde{q}_{t,1}, \ldots, \widetilde{q}_{t,k}\}$ and $\widetilde{A}_t = \{\widetilde{a}_{t,1}, \ldots, \widetilde{a}_{t,k}\}$. Update distribution $D$:

$$D_t \leftarrow \arg\min_{D \in \mathcal{D}} \mathcal{L}\left(D_{t-1}, \widetilde{Q}_{1:t}, \widetilde{A}_{1:t}\right)$$

    where $\widetilde{Q}_{1:t} = \bigcup_{i=1}^{t} \widetilde{Q}_i$ and $\widetilde{A}_{1:t} = \bigcup_{i=1}^{t} \widetilde{A}_i$.

**end**

Output $H\left(\{D_t\}_{t=0}^{T}\right)$ where $H$ is some function over all distributions $D_t$ (such as the average)

---

Finally, we note that in addition to the loss function $\mathcal{L}$, a key component that differentiates algorithms under this framework is the distributional family $\mathcal{D}$ that the output of each algorithm belongs to. We refer readers to Appendix A.2, where we describe in more detail how existing algorithms fit into our general framework under different choices of $\mathcal{L}$ and $\mathcal{D}$.

### 3.1 Privacy analysis

We present the privacy analysis of the Adaptive Measurements framework while assuming that the exponential and Gaussian mechanism are used for the private *sample* and noisy *measure* steps respectively. More specifically, suppose that we (1) sample $k$ queries using the *exponential mechanism* with the score function:

$$\Pr\left[\widetilde{q}_{t,i} = q\right] \propto \exp\left(\alpha \varepsilon_0 n |q(P) - q(D_{t-1})|\right)$$

and (2) measure the answer to each query by adding Gaussian noise

$$z_{t,i} \sim \mathcal{N}\left(0, \left(\frac{1}{n(1-\alpha)\varepsilon_0}\right)^2\right).$$

Letting $\varepsilon_0 = \sqrt{\frac{2\rho}{T\left(\alpha^2 + (1-\alpha)^2\right)}}$ and $\alpha \in (0,1)$ be a privacy allocation hyperparameter (higher values of $\alpha$ allocate more privacy budget to the exponential mechanism), we present the following theorem:

**Theorem 1.** *When run with privacy parameter $\rho$,* Adaptive Measurements *satisfies $\rho$-zCDP. Moreover for all $\delta > 0$,* Adaptive Measurements *satisfies$(\varepsilon(\delta), \delta)$-differential privacy, where $\varepsilon(\delta) \leq \rho + 2\sqrt{\rho \log(1/\delta)}$.*

*Proof sketch.* Fix $T \geq 1$ and $\alpha \in (0,1)$. (i) At each iteration $t \in [T]$, Adaptive Measurements runs the exponential mechanism $k$ times with parameter $2\alpha\varepsilon_0$, which satisfies $\frac{k}{8}\left(2\alpha\varepsilon_0\right)^2 = \frac{k}{2}\left(\alpha\varepsilon_0\right)^2$-*zCDP* [12], and the Gaussian mechanism $k$ times with parameter $(1-\alpha)\varepsilon_0$, which satisfies $\frac{k}{2}\left[(1-\alpha)\varepsilon_0\right]^2$-*zCDP* [10]. (ii) using the composition theorem for concentrated differential privacy [10], Adaptive Measurements satisfies $\frac{kT}{2}\left[\alpha^2 + (1-\alpha)^2\right]\varepsilon_0^2$-*zCDP* after $T$ iterations. (iii) Setting $\varepsilon_0 = \sqrt{\frac{2\rho}{kT\left(\alpha^2 + (1-\alpha)^2\right)}}$, we conclude that Adaptive Measurements satisfies $\rho$-*zCDP*, which in turn implies $\left(\rho + 2\sqrt{\rho \log(1/\delta)}, \delta\right)$-*differential privacy* for all $\delta > 0$ [10].

## 4  Overcoming Computational Intractability with Generative Networks

We introduce GEM (Generative Networks with the Exponential Mechanism), which optimizes over past queries to improve accuracy by training a generator network $G_\theta$ to implicitly learn a distribution

of the data domain, where $G_\theta$ can be any neural network parametrized by weights $\theta$. As a result, our method GEM can compactly represent a distribution for any data domain while enabling fast, gradient-based optimization via auto-differentiation frameworks [32, 2].

Concretely, $G_\theta$ takes random Gaussian noise vectors $z$ as input and outputs a representation $G_\theta(z)$ of a product distribution over the data domain. Specifically, this product distribution representation takes the form of a $d'$-dimensional probability vector $G_\theta(z) \in [0, 1]^{d'}$, where $d'$ is the dimension of the data in one-hot encoding and each coordinate $G_\theta(z)_j$ corresponds to the marginal probability of a categorical variable taking on a specific value. To obtain this probability vector, we choose softmax as the activation function for the output layer in $G_\theta$. Therefore, for any fixed weights $\theta$, $G_\theta$ defines a distribution over $\mathcal{X}$ through the generative process that draws a random $z \sim \mathcal{N}(0, \sigma^2 I)$ and then outputs random $x$ drawn from the product distribution $G_\theta(z)$. We will denote this distribution as $P_\theta$.

To define the loss function for GEM, we require that it be differentiable so that we can use gradient-based methods to optimize $G_\theta$. Therefore, we need to obtain a differentiable variant of $q$. Recall first that a query is defined by some predicate function $\phi : \mathcal{X} \to \{0, 1\}$ over the data domain $\mathcal{X}$ that evaluates over a single row $x \in \mathcal{X}$. We observe then that one can extend any statistical query $q$ to be a function that maps a distribution $P_\theta$ over $\mathcal{X}$ to a value in $[0, 1]$:

$$q(P_\theta) = \mathbb{E}_{x \sim P_\theta}[\phi(x)] = \sum_{x \in \mathcal{X}} \phi(x) P_\theta(x) \tag{2}$$

Note that any statistical query $q$ is then differentiable w.r.t. $\theta$:

$$\nabla_\theta [q(P_\theta)] = \sum_{x \in \mathcal{X}} \nabla_\theta P_\theta(x) \phi(x) = \mathbb{E}_{\mathbf{z} \sim N(0, I_k)} \left[ \sum_{x \in \mathcal{X}} \phi(x) \nabla_\theta \left[ \frac{1}{k} \sum_i^k \prod_{j=1}^{d'} (G_\theta(z_i)_j)^{x_j} \right] \right]$$

and we can compute stochastic gradients of $q$ w.r.t. $\theta$ with random noise samples $z$. This also allows us to derive a differentiable loss function in the Adaptive Measurements framework. In each round $t$, given a set of selected queries $\widetilde{Q}_{1:t}$ and their noisy measurements $\widetilde{A}_{1:t}$, GEM minimizes the following $\ell_1$-loss:

$$\mathcal{L}^{\mathsf{GEM}}\left(\theta, \widetilde{Q}_{1:t}, \widetilde{A}_{1:t}\right) = \sum_{i=1}^{t} |\widetilde{q}_i(P_\theta) - \widetilde{a}_i|. \tag{3}$$

where $\widetilde{q}_i \in \widetilde{Q}_{1:t}$ and $\widetilde{a}_i \in \widetilde{A}_{1:t}$.

In general, we can optimize $\mathcal{L}^{\mathsf{GEM}}$ by running stochastic (sub-)gradient descent. However, we remark that gradient computation can be expensive since obtaining a low-variance gradient estimate often requires calculating $\nabla_\theta P_\theta(x)$ for a large number of $x$. In Appendix B, we include the stochastic gradient derivation for GEM and briefly discuss how an alternative approach from reinforcement learning.

For many query classes, however, there exists some closed-form, differentiable function surrogate to (3) that evaluates $q(G_\theta(z))$ directly without operating over all $x \in \mathcal{X}$. Concretely, we say that for certain query classes, there exists some representation $f_q : \Delta(\mathcal{X}) \to [0, 1]$ for $q$ that operates in the probability space of $\mathcal{X}$ and is also differentiable.

In this work, we implement GEM to answer $k$-way marginal queries, which have been one of the most important query classes for the query release literature [22, 40, 19, 26] and provides a differentiable form when extended to be a function over distributions. In particular, we show that $k$-way marginals can be rewritten as product queries (which are differentiable).

**Definition 4** (Product query). *Let $p \in \mathbb{R}^{d'}$ be a representation of a dataset (in the one-hot encoded space), and let $S \subseteq [d']$ be some subset of dimensions of $p$. Then we define a product query $f_S$ as*

$$f_S(p) = \prod_{j \in S} p_j \tag{4}$$

A $k$-way marginal query $\phi$ can then be rewritten as (4), where $p = G_\theta(z)$ and $S$ is the subset of dimensions corresponding to the attributes $A$ and target values $y$ that are specified by $\phi$ (Definition 1). Thus, we can write any marginal query as $\prod_{j \in S} G_\theta(z)_j$, which is differentiable w.r.t. $G_\theta$ (and therefore differentiable w.r.t weights $\theta$ by chain rule). Gradient-based optimization techniques can then be used to solve (3); the exact details of our implementation can be found in Appendix B.

# 5 Maximum-Entropy Projection Algorithm

Next, we propose PEP (Private Entropy Projection) under the framework Adaptive Measurements. Similar to MWEM, PEP employs the *maximum entropy* principle, which also recovers a synthetic data distribution in the exponential family. However, since PEP adaptively assigns weights to past queries and measurements, it has faster convergence and better accuracy than MWEM. PEP's loss function in Adaptive Measurements can be derived through a constrained maximum entropy optimization problem. At each round $t$, given a set of selected queries $\widetilde{Q}_{1:t}$ and their noisy measurements $\widetilde{A}_{1:t}$, the constrained optimization requires that the synthetic data $D_t$ satisfies $\gamma$-accuracy with respect to the noisy measurements for all the selected queries in $\widetilde{Q}_{1:t}$. Then among the set of feasible distributions that satisfy accuracy constraints, PEP selects the distribution with maximum entropy, leading to the following regularized constraint optimization problem:

$$\text{minimize:} \quad \sum_{x \in \mathcal{X}} D(x) \log \left( D(x) \right) \tag{5}$$

$$\text{subject to:} \quad \forall_{i \in [t]} \quad |\widetilde{a}_i - \widetilde{q}_i(D)| \leq \gamma, \quad \sum_{x \in \mathcal{X}} D(x) = 1$$

We can ignore the constraint that $\forall_{x \in \mathcal{X}} D(x) \geq 0$, because it will be satisfied automatically.

The solution of (5) is an exponentially weighted distribution parameterized by the dual variables $\lambda_1, \ldots, \lambda_t$ corresponding to the $t$ constraints. Therefore, if we solve the dual problem of (5) in terms of the dual variables $\lambda_1, \ldots, \lambda_t$, then the distribution that minimizes (5) is given by $D_t(x) \propto \exp\left(\sum_{i=1}^{t} \lambda_i \widetilde{q}_i(x)\right)$. (Note that MWEM simply sets $\lambda_i = \tilde{a}_i - \tilde{q}_i(D_{t-1})$ without any optimization.) Given that the set of distributions is parameterized by the variables $\lambda_1, \ldots, \lambda_t$, the constrained optimization is then equivalent to minimizing the following exponential loss function:

$$\mathcal{L}^{\text{PEP}} \left(\lambda, \widetilde{Q}_{1:t}, \widetilde{A}_{1:t}\right) = \log \left( \sum_{x \in \mathcal{X}} \exp \left( \sum_{i=1}^{t} \lambda_i \left( \widetilde{q}_i(x) - \widetilde{a}_i \right) \right) \right) + \gamma \|\lambda\|_1$$

Since PEP requires solving this constrained optimization problem on each round, we give an efficient iterative algorithm for solving (5). We defer the details of the algorithm to the Appendix C.

# 6 Extending to the *public-data-assisted* setting

Incorporating prior information from public data has shown to be a promising avenue for private query release [6, 26]. Therefore, we extend GEM to the problem of *public-data-assisted private (PAP) query release* [6] in which differentially private algorithms have access to public data. Concretely, we adapt GEM to utilize public data by initializing $D_0$ to a distribution over the public dataset. However, because in GEM, we implicitly model any given distribution using a generator $G$, we must first train without privacy (i.e., without using the exponential and Gaussian mechanisms) a generator $G_{\text{pub}}$, to minimize the $\ell_1$-error over some set of queries $\hat{Q}$. Note that in most cases, we can simply let $\hat{Q} = Q$ where $Q$ is the collection of statistical queries we wish to answer privately. GEM$^{\text{Pub}}$ then initializes $G_0$ to $G_{\text{pub}}$ and proceeds with the rest of the GEM algorithm.

## 6.1 Overcoming limitations of PMW$^{\text{Pub}}$.

We describe the limitations of PMW$^{\text{Pub}}$ by providing two example categories of public data that it fails to use effectively. We then describe how GEM$^{\text{Pub}}$ overcomes such limitations in both scenarios.

**Public data with insufficient support.** We first discuss the case in which the public dataset has an insufficient support, which in this context means the support has high *best-mixture-error* [26]. Given some support $S \subseteq \mathcal{X}$, the *best-mixture-error* can be defined as

$$\min_{\mu \in \Delta(S)} \max_{q \in Q} \left| q(D) - \sum_{x \in S} \mu_x q(x) \right|$$

where $\mu \in \Delta(S)$ is a distribution over the set $S$ with $\mu(x) \geq 0$ for all $x \in S$ and $\sum_{x \in S} \mu(x) = 1$.

In other words, the *best-mixture-error* approximates the lowest possible max error that can be achieved by reweighting some support, which in this case means PMW$^{\mathsf{Pub}}$ cannot achieve max errors lower that this value. While Liu et al. [26] offer a solution for filtering out poor public datasets ahead of time using a small portion of the privacy budget, PMW$^{\mathsf{Pub}}$ cannot be run effectively if no other suitable public datasets exist. GEM$^{\mathsf{Pub}}$ however avoids this issue altogether because unlike MWEM (and therefore PMW$^{\mathsf{Pub}}$), which cannot be run without restricting the size of $\mathcal{D}$, GEM$^{\mathsf{Pub}}$ can utilize public data without restricting the distributional family it can represent (since both GEM and GEM$^{\mathsf{Pub}}$ compactly parametrize any distribution using a neural network).

**Public data with incomplete data domains.** Next we consider the case in which the public dataset only has data for a subset of the attributes found in the private dataset. We note that as presented in Liu et al. [26], PMW$^{\mathsf{Pub}}$ cannot handle this scenario. One possible solution is to augment the public data distribution by assuming a uniform distribution over all remaining attributes missing in the public dataset. However, while this option may work in cases where only a few attributes are missing, the missing support grows exponentially in the dimension of the missing attributes. In contrast, GEM$^{\mathsf{Pub}}$ can still make use of such public data. In particular, we can pretrain a generator $G$ on queries over just the attributes found in the public dataset. Again, GEM$^{\mathsf{Pub}}$ avoids the computational intractability of PMW$^{\mathsf{Pub}}$ in this setting since it parametrizes its output distribution with $G$.

# 7 Empirical Evaluation

In this section, we empirically evaluate GEM and PEP against baseline methods on the ACS [34] and ADULT [13] datasets in both the standard[3] and *public-data-assisted* settings.

**Data.** To evaluate our methods, we construct public and private datasets from the ACS and ADULT datasets by following the preprocessing steps outlined in Liu et al. [26]. For the ACS, we use 2018 data for the state of Pennsylvania (PA-18) as the private dataset. For the public dataset, we select 2010 data for Pennsylvania (PA-10) and 2018 data for California (CA-18). In our experiments on the ADULT dataset, private and public datasets are sampled from the complete dataset (using a 90-10 split). In addition, we construct low-dimensional versions of both datasets, which we denote as ACS (reduced) and ADULT (reduced), in order to evaluate PEP and MWEM.

**Baselines.** We compare our algorithms to the strongest performing baselines in both low and high-dimensional settings, presenting results for MWEM, DualQuery, and RAP$^{\mathsf{softmax}}$ in the standard setting[45] and PMW$^{\mathsf{Pub}}$ in the *public-data-assisted* setting.

**Experimental details.** To present a fair comparison, we implement all algorithms using the privacy mechanisms and *zCDP* composition described in Section 3.1. To implement GEM for $k$-way marginals, we select a simple multilayer perceptron for $G_\theta$. Our implementations of MWEM and PMW$^{\mathsf{Pub}}$ output the last iterate $D_t$ instead of the average and apply the multiplicative weights update rule using past queries according to the pseudocode described in Liu et al. [26]. We report the best performing 5-run average across hyperparameter choices (see Tables 1, 2, 3, 4, and 5 in Appendix D.1) for each algorithm.

## 7.1 Results

**Standard setting.** In Figure 1, we observe that in low-dimensional settings, PEP and GEM consistently achieve strong performance compared to the baseline methods. While MWEM and PEP are similar in nature, PEP outperforms MWEM on both datasets across all privacy budgets except

---

[3]We refer readers to Appendix D.7 where we include an empirical evaluation on versions of the ADULT and LOANS [13] datasets used in other related private query release works [28, 40, 5].

[4]Having consulted McKenna et al. [28], we concluded that running HDMM is infeasible for our experiments, since it generally cannot handle a data domain with size larger than $10^9$. See Appendix D.5 for more details.

[5]Because RAP performs poorly relative to the other methods in our experiments, plotting its performance would make visually comparing the other methods difficult. Thus, we exclude it from Figure 1 and refer readers to Appendix D.3, where we present failure cases for RAP and compare it to RAP$^{\mathsf{softmax}}$.

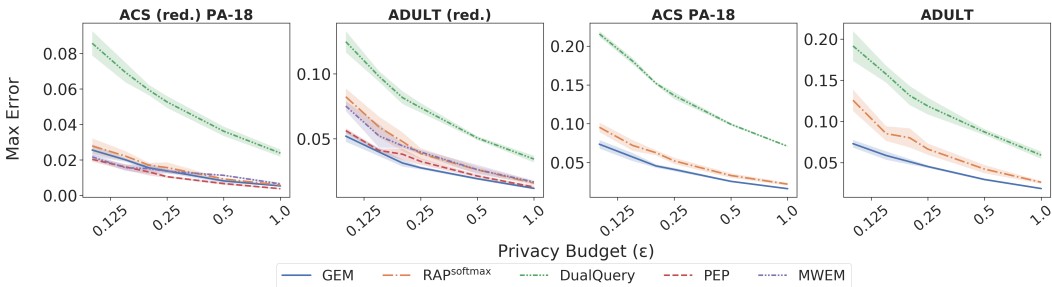

Figure 1: Max error for 3-way marginals evaluated on ADULT and ACS PA-18 using privacy budgets $\varepsilon \in \{0.1, 0.15, 0.2, 0.25, 0.5, 1\}$ and $\delta = \frac{1}{n^2}$. The *x-axis* uses a logarithmic scale. We evaluate using the following workload sizes: ACS (reduced) PA-18: 455; ADULT (reduced): 35; ACS PA-18: 4096; ADULT: 286. Results are averaged over 5 runs, and error bars represent one standard error.

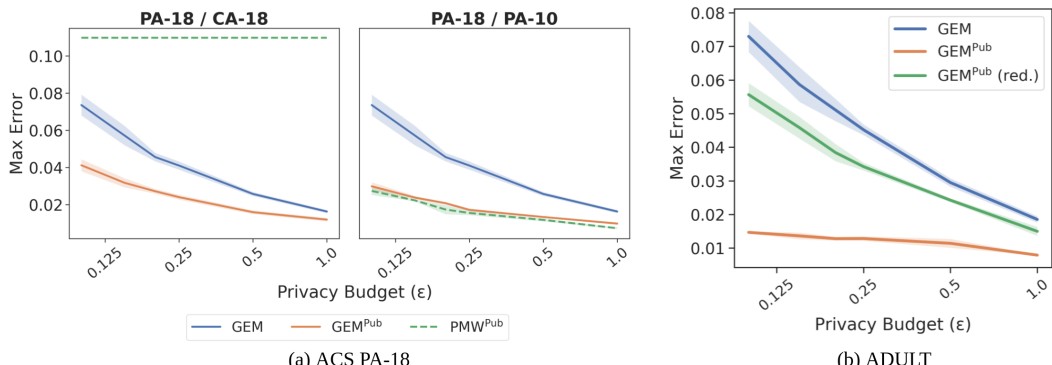

Figure 2: Max error for 3-way marginals with privacy budgets $\varepsilon \in \{0.1, 0.15, 0.2, 0.25, 0.5, 1\}$ and $\delta = \frac{1}{n^2}$. The *x-axis* uses a logarithmic scale. Results are averaged over 5 runs, and error bars represent one standard error. **(a)** ACS PA-18 (workloads = 4096). We evaluate public-data-assisted algorithms with the following public datasets: *Left:* 2018 California (CA-18); *Right:* 2010 Pennsylvania (PA-10). **(b)** ADULT (workloads = 286). We evaluate GEM using both the complete public data (GEM$^{\mathsf{Pub}}$) and a reduced version that has fewer attributes (GEM$^{\mathsf{Pub}}$ (reduced)).

$\varepsilon \in \{0.1, 0.15\}$ on ACS (reduced), where the two algorithms perform similarly. In addition, both PEP and GEM outperform RAP$^{\mathsf{softmax}}$. Moving on to the more realistic setting in which the data dimension is high, we again observe that GEM outperforms RAP$^{\mathsf{softmax}}$ on both datasets.

***Public-data-assisted* setting.** To evaluate the query release algorithm in the public-data-assisted setting, we present the three following categories of public data:

*Public data with sufficient support.* To evaluate our methods when the public dataset for ACS PA-18 has low *best-mixture-error*, we consider the public dataset ACS PA-10. We observe in Figure 1 that GEM$^{\mathsf{Pub}}$ performs similarly to PMW$^{\mathsf{Pub}}$, with both outperforming GEM (without public data).

*Public data with insufficient support.* In Figure 2a, we present CA-18 as an example of this failure case in which the *best-mixture-error* is over $10\%$, and so for any privacy budget, PMW$^{\mathsf{Pub}}$ cannot achieve max errors lower that this value. However, for the reasons described in Section 6.1, GEM$^{\mathsf{Pub}}$ is not restricted by *best-mixture-error* and significantly outperforms GEM (without public data) when using either public dataset.

*Public data with incomplete data domains.* To simulate this setting, we construct a reduced version of the public dataset in which we keep only 7 out of 13 attributes in ADULT. In this case, 6 attributes are missing, and so assuming a uniform distribution over the missing attributes would cause the dimension of the approximating distribution $D$ to grow from $\approx 4.4 \times 10^3$ to a $\approx 3.2 \times 10^9$. PMW$^{\mathsf{Pub}}$ would be computationally infeasible to run in this case. To evaluate GEM$^{\mathsf{Pub}}$, we pretrain the generator $G$ using all 3-way marginals on both the complete and reduced versions of the public

dataset and then finetune on the private dataset (we denote these two finetuned networks as $\mathsf{GEM}^{\mathsf{Pub}}$ and $\mathsf{GEM}^{\mathsf{Pub}}$ (reduced) respectively). We present results in Figure 2b. Given that the public and private datasets are sampled from the same distribution, $\mathsf{GEM}^{\mathsf{Pub}}$ unsurprisingly performs extremely well. However, despite only being pretrained on a small fraction of all 3-way marginal queries ($\approx 20k$ out $334k$), $\mathsf{GEM}^{\mathsf{Pub}}$ (reduced) is still able to improve upon the performance of $\mathsf{GEM}$ and achieve lower max error for all privacy budgets.

## 8 Conclusion

In this work, we present a framework that unifies a long line of iterative private query release algorithms by reducing each method to a choice of some distributional family $\mathcal{D}$ and loss function $\mathcal{L}$. We then develop two new algorithms, $\mathsf{PEP}$ and $\mathsf{GEM}$, that outperform existing query release algorithms. In particular, we empirically validate that $\mathsf{GEM}$ performs very strongly in high dimensional settings (both with and without public data). We note that we chose a rather simple neural network architecture for $\mathsf{GEM}$, and so for future work, we hope to develop architectures more tailored to our problem. Furthermore, we hope to extend our algorithms to other query classes, including mixed query classes and convex minimization problems [37].

**Limitations.** While our empirical study is structured based on past work in the literature, real-world deployment of differential privacy may be more nuanced. For example, though $\mathsf{GEM}$ achieves strong performance on the ACS dataset in our experiments, we (and possibly the U.S. Census Bureau itself) do not know yet what the end goals of a differentially private ACS release should be. Given different constraints or evaluation metrics, it is possible for another technique to be more suitable.

**Broader impacts.** We note that a possible consequence of privacy preserving methods is their potential to bring about unfair outcomes. Fioretto et al. [18], for example, show that decisions based on DP analyses may disproportionately impact certain groups. Because the U.S. Census Bureau has plans to adopt differential privacy for ACS releases after 2025 [23], the ACS dataset serves as the main test bed for our algorithms. We believe it is important to ensure that fairness is preserved for downstream tasks that use DP synthetic dataset, since the ACS provides statistics that are critical to many important decision processes (e.g., Title I grant allocations [1]).

## Acknowledgments

ZSW is supported by NSF grant SCC-1952085, Carnegie Mellon CyLab's Secure and Private IoT Initiative, and a Google Faculty Research Award.

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
