# A   Adaptive Measurements

## A.1   $k$-way marginals sensitivity

Typically, iterative private query release algorithms assume that the query class $Q$ contains sensitivity 1 queries [21, 19, 40, 5, 26]. However, recall that a $k$-way marginal query is defined by a subset of features $S \subseteq [d]$ and a target value $y \in \prod_{i \in S} \mathcal{X}_i$ (Definition 1). Given a feature set $S$ (with $|S| = k$), we can define a workload $W_S$ as the set of queries defined over the features in $S$.

$$W_S = \left\{ q_{S,y} : y \in \prod_{i \in S} \mathcal{X}_i \right\}$$

Then for any dataset $D$, the workload's answer is given by $W_S(D) = \{q_{S,y}(D)\}_{y \in \prod_{i \in S} \mathcal{X}_i}$, where $W_S$ has $\ell_2$-sensitivity equal to $\sqrt{2}$. Therefore to achieve more efficient privacy accounting for $k$-way marginals in Adaptive Measurements, we can use the exponential mechanism to select an entire workload $\hat{W}_S$ that contains the max error query and then obtain measurements for *all* queries in $\hat{W}_S$ using the Gaussian mechanism, adding noise

$$z \sim \mathcal{N}\left(0, \left(\frac{\sqrt{2}}{n(1-\alpha)\varepsilon_0}\right)^2\right).$$

to each query in $W_S$.

In appendix D.4, we show that using this marginal trick significantly improves the performance of GEM and therefore recommend this type of privacy accounting when designing query release algorithms for $k$-way marginal queries.

## A.2   Choices of loss functions and distributional families

We provide additional details about each iterative algorithm, including the loss function $\mathcal{L}$ and and distributional family $\mathcal{D}$ (under the Adaptive Measurements framework).

MWEM **from Hardt et al. [22]**   The traditional MWEM algorithm samples one query each round, where after $t$ rounds, the set of queries/measurements is $\widetilde{Q}_t = \{\widetilde{q}_1, \ldots, \widetilde{q}_t\}, \widetilde{A}_t = \{\widetilde{a}_1, \ldots, \widetilde{a}_t\}$. Let the $t - 1$ previous solutions be $D_1, \ldots D_{t-1}$. Then MWEM solves an entropy regularized problem in which it finds $D_t$ that minimizes the following loss:

$$\mathcal{L}^{\mathsf{MWEM}}(D, \widetilde{Q}_{1:t}, \widetilde{A}_{1:t}) = \sum_{i=1}^{t} \sum_{x \in \mathcal{X}} D(x)\widetilde{q}_i(x)\left(\widetilde{a}_i - \widetilde{q}_i(D_{i-1})\right) + \sum_{x \in \mathcal{X}} D(x) \log D(x)$$

We can show that if $D_t = \arg\min_{D \in \Delta(\mathcal{X})} \mathcal{L}^{\mathsf{MWEM}}(D, \widetilde{Q}_t, \widetilde{A}_t)$ then $D_t$ evaluates to $D_t(x) \propto \exp\left(-\sum_{i=1}^{t} \widetilde{q}_i(x)(\widetilde{a}_i - \widetilde{q}_i(D_{i-1}))\right)$ which is the exactly the distribution computed by MWEM. See A.3 for derivation. We note that MWEM explicitly maintains (and outputs) a distribution $D \in \mathcal{D}$ where $\mathcal{D}$ includes all distributions over the data domain $\mathcal{X}$, making it computationally intractable for high-dimension settings.

DualQuery **from Gaboardi et al. [19]**   DualQuery is a special case of the Adaptive Measurements framework in which the measurement step is skipped (abusing notation, we say $\alpha = 1$). Over all iterations of the algorithm, DualQuery keeps track of a probability distribution over the set of queries $Q$ via multiplicative weights, which we denote here by $\mathcal{Q}_t \in \Delta(Q)$. On round $t$, DualQuery samples $s$ queries ($\widetilde{Q}_t = \{\widetilde{q}_{t,1}, \ldots \widetilde{q}_{t,s}\}$) from $\mathcal{Q}_t$ and outputs $D_t$ that minimizes the the following loss function:

$$\mathcal{L}^{\mathsf{DualQuery}}(D, \widetilde{Q}_t) = \sum_{i=1}^{s} \widetilde{q}_{t,i}(D)$$

The optimization problem for $\mathcal{L}^{\mathsf{DualQuery}}(D, \widetilde{Q}_t)$ is NP-hard. However, the algorithm encodes the problem as a mix-integer-program (MIP) and takes advantage of available fast solvers. The final output of DualQuery is the average $\frac{1}{T}\sum_{t=1}^{T} D_t$, which we note implicitly describes some empirical distribution over $\mathcal{X}$.

**FEM from Vietri et al. [40]**  The algorithm FEM follows a follow the perturbed leader strategy. As with MWEM, the algorithm FEM samples one query each round using the exponential mechanism, so that the set of queries in round $t$ is $\widetilde{Q}_t = \{\widetilde{q}_1, \ldots, \widetilde{q}_t\}$. Then on round $t$, FEM chooses the next distribution by solving:

$$\mathcal{L}^{\mathsf{FEM}}(D, \widetilde{Q}_{1:t}) = \sum_{i=1}^{t} \widetilde{q}_t(D) + \mathbb{E}_{x \sim D, \eta \sim \mathrm{Exp}(\sigma)^d}\left(\langle x, \eta \rangle\right)$$

Similar to DualQuery, the optimization problem for $\mathcal{L}^{\mathsf{FEM}}$ also involves solving an NP-hard problem. Additionally, because the function $\mathcal{L}^{\mathsf{FEM}}$ does not have a closed form due to the expectation term, FEM follows a sampling strategy to approximate the optimal solution. On each round, FEM generates $s$ samples, where each sample is obtained in the following way: Sample a noise vector $\eta \sim \mathrm{Exp}(\sigma)^d$ from the exponential distribution and use a MIP to solve $x_{t,i} \leftarrow \arg\min_{x \in \mathcal{X}} \sum_{i=1}^{t} \widetilde{q}_t(D) + \langle x, \eta \rangle$ for all $i \in [s]$. Finally, the output on round $t$ is the empirical distribution derived from the $s$ samples: $D_t = \{x_{t,1}, \ldots, x_{t,s}\}$. The final output is the average $\frac{1}{T}\sum_{t=1}^{T} D_t$.

**RAP$^{\mathsf{softmax}}$ adapted from Aydore et al. [5]**  At iteration $t$, RAP$^{\mathsf{softmax}}$ solves the following optimization problem:

$$\mathcal{L}^{\mathsf{RAP}}(D, \widetilde{Q}_{1:t}, \widetilde{A}_{1:t}) = \sum_{i,j}\left(\widetilde{q}_{i,j}(D) - \widetilde{a}_{i,j}\right)^2$$

As stated in Section A, we apply the softmax function such that RAP$^{\mathsf{softmax}}$ outputs a synthetic dataset drawn from some probabilistic family of distributions $\mathcal{D} = \left\{\sigma(M) | M \in \mathbb{R}^{n' \times d}\right\}$.

### A.3  MWEM **update**

Given the loss function:

$$\mathcal{L}^{\mathsf{MWEM}}(D, \widetilde{Q}_t, \widetilde{A}_t) = \sum_{i=1}^{t}\sum_{x \in \mathcal{X}} D(x)\widetilde{q}_i(x)\left(\widetilde{a}_i - \widetilde{q}_i(D_{i-1})\right) + \sum_{x \in \mathcal{X}} D(x)\log\left(D(x)\right) \quad (6)$$

The optimization problem becomes $D_t = \arg\min_{D \in \Delta(\mathcal{X})} \mathcal{L}^{\mathrm{mwem}}(D, \widetilde{Q}_t, \widetilde{A}_t)$. The solution $D$ is some distribution, which we can express as a constraint $\sum_{x \in \mathcal{X}} D(x) = 1$. Therefore, this problem is a constrained optimization problem. To show that (6) is the MWEM's true loss function, we can write down the Lagrangian as:

$$\mathcal{L} = \sum_{i=1}^{t}\sum_{x \in \mathcal{X}} D(x)\widetilde{q}_i(x)\left(\widetilde{a}_i - \widetilde{q}_i(D_{i-1})\right) + \sum_{x \in \mathcal{X}} D(x)\log\left(D(x)\right) + \lambda\left(\sum_{x \in \mathcal{X}} D(x) - 1\right)$$

Taking partial derivative with respect to $D(x)$:

$$\frac{\partial \mathcal{L}}{\partial D(x)} = \sum_{i=1}^{t} \widetilde{q}_i(x)\left(\widetilde{a}_i - \widetilde{q}_i(D_{i-1})\right) + (1 + \log D(x)) + \lambda$$

Setting $\frac{\partial \mathcal{L}}{\partial D(x)} = 0$ and solving for $D(x)$:

$$D(x) = \exp\left(-1 - \lambda - \sum_{i=1}^{t} \widetilde{q}_i(x)\left(\widetilde{a}_i - \widetilde{q}_i(D_{i-1})\right)\right)$$

Finally, the value of $\lambda$ is set such that $D$ is a probability distribution:

$$D(x) = \frac{\exp\left(-\sum_{i=1}^{t} \widetilde{q}_i(x)\left(\widetilde{a}_i - \widetilde{q}_i(D_{i-1})\right)\right)}{\sum_{x \in \mathcal{X}} \exp\left(-\sum_{i=1}^{t} \widetilde{q}_i(x)\left(\widetilde{a}_i - \widetilde{q}_i(D_{i-1})\right)\right)}$$

This concludes the derivation of MWEM loss function.

## B GEM

We show the exact details of GEM in Algorithms 2 and 3. Note that given a vector of queries $Q_t = \langle q_1, \ldots, q_t \rangle$, we define $f_{Q_t}(\cdot) = \langle f_{q_1}(\cdot), \ldots, f_{q_t}(\cdot) \rangle$.

---

**Algorithm 2:** GEM

---

**Input:** Private dataset $P$, set of differentiable queries $Q$
**Parameters**: privacy parameter $\rho$, number of iterations $T$, privacy weighting parameter $\alpha$, batch size $B$, stopping threshold $\gamma$
Initialize generator network $G_0$
Let $\varepsilon_0 = \sqrt{\frac{2\rho}{T(\alpha^2 + (1-\alpha)^2)}}$
**for** $t = 1 \ldots T$ **do**

    **Sample:** Sample $\mathbf{z} = \langle z_1 \ldots z_B \rangle \sim \mathcal{N}(0, I_B)$
    Choose $\widetilde{q}_t$ using the *exponential mechanism* with score

$$\Pr[q_t = q] \propto \exp\left(\frac{\alpha \varepsilon_0 n}{2} |q(P) - q(G_{t-1}(\mathbf{z}))|\right)$$

    **Measure:** Let $\widetilde{a}_t = \widetilde{q}_t(P) + \mathcal{N}\left(0, \left(\frac{1}{n(1-\alpha)\varepsilon_0}\right)^2\right)$
    **Update:** $G_t = \text{GEM-UPDATE}(G_{t-1}, Q_t, \widetilde{\mathbf{a}}_\mathbf{t}, \gamma)$ where $Q_t = \langle \widetilde{q}_1, \ldots, \widetilde{q}_t \rangle$ and $\widetilde{\mathbf{a}}_\mathbf{t} = \langle \widetilde{a}_1, \ldots, \widetilde{a}_t \rangle$

**end**

Let $\theta_{out} = \text{EMA}\left(\{\theta_j\}_{j=\frac{T}{2}}^T\right)$ where $\theta_j$ parameterizes $G_j$
Let $G_{out}$ be the generator parameterized by $\theta_{out}$
Output $G_{out}(\mathbf{z})$

---

**Algorithm 3:** GEM-UPDATE

---

**Input:** Generator $G$ parameterized by $\theta$, queries $Q$, noisy measurements $\widetilde{\mathbf{a}}$, stopping threshold $\gamma$
**Parameters:** max iterations $T_{\max}$, batch size $B$
Sample $\mathbf{z} = \langle z_1 \ldots z_B \rangle \sim \mathcal{N}(0, I_B)$
Let $\mathbf{c} = \widetilde{\mathbf{a}} - \frac{1}{B}\sum_{j=1}^B f_Q(G(z_j))$ be errors over queries $Q$
Let $i = 0$
**while** $i < T_{max}$ *and* $\|\mathbf{c}\|_\infty \geq \gamma$ **do**

    Let $J = \{j \mid |c_j| \geq \gamma\}$
    Update $G$ to minimize the loss function with the stochastic gradient $\nabla_\theta \frac{1}{|J|} \sum_{j \in J} |c_{ij}|$
    Sample $\mathbf{z} = \langle z_1 \ldots z_B \rangle \sim \mathcal{N}(0, I_B)$
    Let $\mathbf{c} = \widetilde{\mathbf{a}} - \frac{1}{B}\sum_{j=1}^B f_Q(G(z_j))$
    Let $i = i + 1$

**end**
**Output:** $G$

---

### B.1 Loss function (for $k$-way marginals) and distributional family

For any $z \in \mathbb{R}$, $G(z)$ outputs a distribution over each attribute, which we can use to calculate the answer to a query via $f_q$. In GEM however, we instead sample a noise vector $\mathbf{z} = \langle z_1 \ldots z_B \rangle$ and calculate the answer to some query $q$ as $\frac{1}{B}\sum_{j=1}^B f_q(G(z_j))$. One way of interpreting the batch size $B$ is to consider each $G(z_j)$ as a unique distribution. In this sense, GEM models $B$ sub-populations that together comprise the overall population of the synthetic dataset. Empirically, we find that our model tends to better capture the distribution of the overall private dataset in this way (Figure 3). Note that for our experiments, we choose $B = 1000$ since it performs well while still achieving good running time. However, this hyperparameter can likely be further increased or tuned (which we leave to future work).

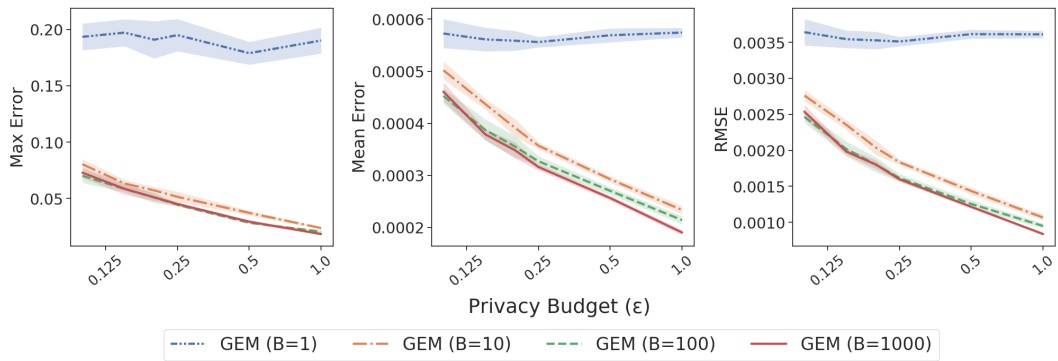

Figure 3: Error comparison of GEM using different batch sizes $B$ on ADULT (workloads=286), evaluated on 3-way marginals with privacy budgets $\varepsilon \in \{0.1, 0.15, 0.2, 0.25, 0.5, 1\}$ and $\delta = \frac{1}{n^2}$. The *x-axis* uses a logarithmic scale. Results are averaged over 5 runs, and error bars represent one standard error.

Therefore, using this notation, GEM outputs then a generator $G \in \mathcal{D}$ by optimizing $\ell_1$-loss at each step $t$ of the Adaptive Measurements framework:

$$\mathcal{L}^{\mathsf{GEM}} \left( G, \widetilde{Q}_{1:t}, \widetilde{A}_{1:t} \right) = \sum_{i=1}^{t} \left| \frac{1}{B} \sum_{j=1}^{B} f_{\widetilde{q}_i} \left( G \left( z_j \right) \right) - \widetilde{a}_i \right| \tag{7}$$

Lastly, we note that we can characterize family of distributions $\mathcal{D} = \{G_\theta(z) | z \sim \mathcal{N}(0, 1)\}$ in GEM by considering it as a class of distributions whose marginal densities are parameterized by $\theta$ and Gaussian noise $z \sim \mathcal{N}(0, 1)$ We remark that such densities can technically be characterized as Boltzmann distributions.

## B.2 Additional implementation details

**EMA output** We observe empirically that the performance of the last generator $G_T$ is often unstable. One possible solution explored previously in the context of privately trained GANs is to output a mixture of samples from a set of generators [8, 30]. In our algorithm GEM, we instead draw inspiration from Yazıcı et al. [43] and output a single generator $G_{out}$ whose weights $\theta_{out}$ are an exponential moving average (EMA) of weights $\theta_t$ obtained from the latter half of training. More concretely, we define $\theta_{out} = \mathsf{EMA} \left( \{\theta_j\}_{j=\frac{T}{2}}^{T} \right)$, where the update rule for EMA is given by $\theta_k^{EMA} = \beta \theta_{k-1}^{EMA} + (1 - \beta)\theta_k$ for some parameter $\beta$.

**Stopping threshold** $\gamma$ To reduce runtime and prevent GEM from overfitting to the sampled queries, we run GEM-UPDATE with some early stopping threshold set to an error tolerance $\gamma$. Empirically, we find that setting $\gamma$ to be half of the max error at each time step $t$. Because sampling the max query using the exponential mechanism provides a noisy approximation of the true max error, we find that using an exponential moving average (with $\beta = 0.5$) of the sampled max errors is a more stable approximation of the true max error. More succinctly, we set $\gamma = \mathsf{EMA}(\{c_i\}_{i=0}^{t})$ where $c_i$ is max error at the beginning of iteration $i$.

**Resampling Gaussian noise.** In our presentation of GEM and in Algorithms 2 and 3, we assume that GEM resamples Gaussian noise $z$. While resampling $z$ encourages GEM to train a generator to output a distribution for any $\mathbf{z} \sim \mathcal{N}(0, I_B)$ for some fixed batch size $B$, we find that fixing the noise vector $\mathbf{z}$ at the beginning of training leads to faster convergence. Moreover in Figure 4, we show that empirically, the performance between whether we resample $z$ at each iteration is not very different. Since resampling $z$ does not induce any benefits to generating synthetic data for the purpose of query release, in which the goal is output a single synthetic dataset or distribution, we run all experiments without resampling $z$. However, we note that it is possible that in other settings, resampling the noise

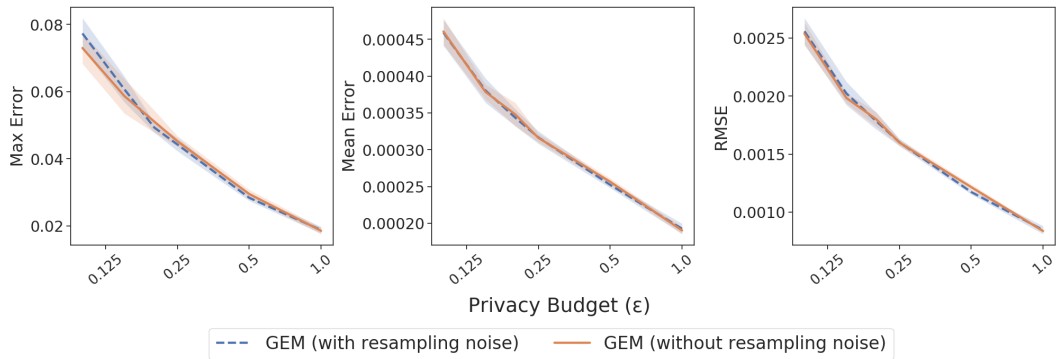

Figure 4: Error comparison of GEM with and without resampling $\mathbf{z}$ at each step on ADULT (workloads=286), evaluated on 3-way marginals with privacy budgets $\varepsilon \in \{0.1, 0.15, 0.2, 0.25, 0.5, 1\}$ and $\delta = \frac{1}{n^2}$. The *x-axis* uses a logarithmic scale. Results are averaged over 5 runs, and error bars represent one standard error.

vector at each step makes more sense and warrants compromising per epoch convergence speed and overall runtime. We leave further investigation to future work.

### B.3  Optimizing over arbitrary query classes

To optimize the loss function for GEM (Equation 3) using gradient-based optimization, we need to have access to the gradient of each $\widetilde{q}_i$ with respect to the input distribution $G_\theta(z)$ for any $z$ (once we compute this gradient, we can then derive the gradient of the loss function with respect to the parameters $\theta$ via chain rule). Given any arbitrary query function $q$, we can rewrite it as (2), which is differentiable w.r.t. $\theta$.

More specifically, for $\mathbf{z} \sim N(0, I_k)$, we write $P_\theta(x) = \mathbb{E}_{\mathbf{z} \sim N(0,I_k)} [P_{\theta,\mathbf{z}}(x)]$, where

$$P_{\theta,\mathbf{z}}(x) = \frac{1}{k} \sum_{i=1}^{k} \prod_{j=1}^{d'} (G_\theta(z_i)_j)^{x_j}$$

Then

$$\begin{aligned}
\nabla_\theta[q(P_\theta)] &= \nabla_\theta \sum_{x \in \mathcal{X}} \phi(x) P_\theta(x) \\
&= \sum_{x \in \mathcal{X}} \phi(x) \nabla_\theta P_\theta(x) \\
&= \sum_{x \in \mathcal{X}} \phi(x) \nabla_\theta \left[ \mathbb{E}_{\mathbf{z} \sim N(0,I_k)} [P_{\theta,\mathbf{z}}(x)] \right] \\
&= \mathbb{E}_{\mathbf{z} \sim N(0,I_k)} \left[ \sum_{x \in \mathcal{X}} \phi(x) \nabla_\theta [P_{\theta,\mathbf{z}}(x)] \right]
\end{aligned}$$

However, while this form allows us to compute the gradient $\nabla_\theta q$ even when $\phi$ itself may not be differentiable w.r.t. $x$ or have a closed form, this method is not computationally feasible when the data domain $\mathcal{X}$ is too large because it requires evaluating $\phi$ on all $x \in \mathcal{X}$. One possible alternative is to construct an unbiased estimator. However, this estimator may suffer from high variance when the number of samples is insufficiently large, inducing a trade-off between computational efficiency and the variance of the estimator.

Incorporating techniques from reinforcement learning, such as the REINFORCE algorithm [41], may serve as alternative ways for optimizing over non-differentiable queries. Specifically, we can approximate (2) in the following way:

$$\nabla_\theta[q(P_\theta)] = \mathbb{E}_{\mathbf{z} \sim N(0, I_k)} \left[ \sum_{x \in \mathcal{X}} \phi(x) \nabla_\theta P_{\theta, \mathbf{z}}(x) \right]$$

$$= \mathbb{E}_{\mathbf{z} \sim N(0, I_k)} \left[ \sum_{x \in \mathcal{X}} \phi(x) \frac{P_{\theta, \mathbf{z}}}{P_{\theta, \mathbf{z}}} \nabla_\theta P_{\theta, \mathbf{z}}(x) \right]$$

$$= \mathbb{E}_{\mathbf{z} \sim N(0, I_k)} \left[ \sum_{x \in \mathcal{X}} \phi(x) P_{\theta, \mathbf{z}}(x) \nabla_\theta \log P_{\theta, \mathbf{z}}(x) \right]$$

We can then approximate this gradient by drawing $m$ samples $\{x_1 \dots x_m\}$ from $P_{\theta, \mathbf{z}}(x)$, giving us

$$\mathbb{E}_{\mathbf{z} \sim N(0, I_k)} \left[ \frac{1}{m} \sum_{i=1}^{k} \phi(x_i) \nabla_\theta \log P_{\theta, \mathbf{z}}(x_i) \right]$$

Further work would be required to investigate whether optimizing such surrogate loss functions is effective when differentiable, closed-form representations of a given query class (e.g., the product query representation of $k$-way marginals) are unavailable.

## C PEP

In the following two sections, we first derive PEP's loss function and and in the next section we derive PEP's update rule (or optimization procedure) that is used to minimize it's loss function.

### C.1 PEP loss function

In this section we derive the loss function that algorithm PEP optimizes over on round $t$. Fixing round $t$, we let $\widetilde{Q}_t \subset Q$ be a subset of queries that were selected using a private mechanism and and let $\widetilde{A}_t$ be the noisy measurements corresponding to $\widetilde{Q}_t$. Then algorithm PEP finds a feasible solution to the problem:

$$
\begin{aligned}
\text{minimize:} \quad & \mathrm{RE}\left(D \parallel U\right) && (8) \\
\text{subject to:} \quad & \forall_{i \in [t]} \quad |\widetilde{a}_i - \widetilde{q}_i(D)| \leq \gamma, \quad \sum_{x \in \mathcal{X}} D(x) = 1
\end{aligned}
$$

The Lagrangian of (8) is :

$$
\mathcal{L} = \mathrm{RE}\left(D \parallel U\right) + \sum_{i=1}^{t} \lambda_i^+ \left(\widetilde{a}_i - \widetilde{q}_i(D) - \gamma\right) + \sum_{i=1}^{t} \lambda_i^- \left(\widetilde{q}_i(D) - \widetilde{a}_i - \gamma\right) + \mu \left(\sum_x D(x) - 1\right)
$$

Let $\lambda \in \mathbb{R}^t$ be a vector with, $\lambda_i = \lambda_i^- - \lambda_i^+$. Then

$$
\mathcal{L} = \mathrm{RE}\left(D \parallel U\right) + \sum_{i=1}^{t} \lambda_i \widetilde{q}_i(D) - \sum_{i=1}^{t} \lambda_i \widetilde{a}_i - \gamma \sum_{i=1}^{t} \left(\lambda_i^+ + \lambda_i^-\right) + \mu \left(\sum_x D(x) - 1\right) \quad (9)
$$

where $\|\lambda\|_1 = \sum_{i=1}^{t} \left(\lambda_i^+ + \lambda_i^-\right)$. Taking the derivative with respect to $D(x)$ and setting to zero, we get:

$$
0 = \frac{\partial \mathcal{L}}{\partial D(x)} = \log\left(\frac{D(x)}{U(x)}\right) + 1 + \sum_{i=1}^{t} \lambda_i \widetilde{q}_i(x) + \mu
$$

Solving for $D(x)$, we get

$$
D(x) = U(x) \exp\left(-\sum_{i=1}^{t} \lambda_i \widetilde{q}_i(x) - \mu - 1\right)
$$

The slack variable $\mu$ must be selected to satisfy the constraint that $\sum_{x \in \mathcal{X}} D(x) = 1$. Therefore have that the solution to (8) is a distribution parameterized by the parameter $\lambda$, such that for any $x \in \mathcal{X}$ we have

$$
D(x) = \frac{U(x)}{Z} \exp\left(-\sum_{i=1}^{t} \lambda_i \widetilde{q}_i(x)\right)
$$

where $Z = \sum_{x \in \mathcal{X}} U(x) \exp\left(-\sum_{i=1}^{t} \lambda_i \widetilde{q}_i(x)\right)$. Plugging into (9), we get

$$\begin{aligned}
\mathcal{L} &= \sum_{x \in \mathcal{X}} D(x) \log\left(\frac{D(x)}{U(x)}\right) + \sum_{i=1}^{t} \lambda_i \widetilde{q}_i(D) - \sum_{i=1}^{t} \lambda_i \widetilde{a}_i - \gamma \|\lambda\|_1 \\
&= \sum_{x \in \mathcal{X}} D(x) \left(-\sum_{i=1}^{t} \lambda_i \widetilde{q}_i(x)\right) - \log(Z) + \sum_{i=1}^{t} \lambda_i \widetilde{q}_i(D) - \sum_{i=1}^{t} \lambda_i \widetilde{a}_i - \gamma \|\lambda\|_1 \\
&= -\sum_{i=1}^{t} \lambda_i \widetilde{q}_i(D) - \log(Z) + \sum_{i=1}^{t} \lambda_i \widetilde{q}_i(D) - \sum_{i=1}^{t} \lambda_i \widetilde{a}_i - \gamma \|\lambda\|_1 \\
&= -\log(Z) + \sum_{i=1}^{t} \lambda_i \widetilde{a}_i - \gamma \|\lambda\|_1 \\
&= -\log\left(\frac{Z}{\exp\left(\sum_{i=1}^{t} \lambda_i \widetilde{a}_i\right)}\right) - \gamma \|\lambda\|_1
\end{aligned}$$

Substituting in for $Z$, we get:

$$\begin{aligned}
\mathcal{L} &= -\log\left(\frac{\sum_{x \in \mathcal{X}} \exp\left(\sum_{i=1}^{t} \lambda_i \widetilde{q}_i(x)\right)}{\exp\left(\sum_{i=1}^{t} \lambda_i \widetilde{a}_i\right)}\right) - \gamma \|\lambda\|_1 \\
&= -\log\left(\sum_{x \in \mathcal{X}} \exp\left(\sum_{i=1}^{t} \lambda_i \left(\widetilde{q}_i(x) - \widetilde{a}_i + \gamma\right)\right)\right) - \gamma \|\lambda\|_1
\end{aligned}$$

Finally, we have that the dual problem of (8) finds a vector $\lambda = (\lambda_1, \dots, \lambda_t)$ that maximizes $\mathcal{L}$. We can write the dual problem as a minimization problem:

$$\mathcal{L}(\lambda) = \min_{\lambda} \log\left(\sum_{x \in \mathcal{X}} \exp\left(\sum_{i=1}^{t} \lambda_i \left(\widetilde{q}_i(x) - \widetilde{a}_i\right)\right)\right) + \gamma \|\lambda\|_1$$

### C.2    PEP optimization using iterative projection

In this section we derive the update rule in algorithm 4. Recall that the ultimate goal is to solve (8). Before we describe the algorithm, we remark that it is possible the constraints in problem 8 cannot be satisfied due to the noise we add to the measurements $\widetilde{a}_1, \dots \widetilde{a}_t$. In principle, $\gamma$ can be chosen to be a high-probability upper bound on the noise, which can be calculated through standard concentration bounds on Gaussian noise. In that case, every constraint $|\widetilde{a}_i - \widetilde{q}_i(D)| \leq \gamma$ can be satisfied and the optimization problem is well defined. However, we note that our algorithm is well defined for every choice of $\gamma \geq 0$. For example, in our experiments we have $\gamma = 0$, and we obtain good empirical results that outperform MWEM. In this section we assume that $\gamma = 0$.

To explain how algorithm 4 converges, we cite an established convergence analysis of adaboost from [35, chapter 7] . Similar to adaboost, Algorithm 2 is running iterative projection, where on each iteration, it projects the distribution to satisfy a single constraint. As shown in [35, chapter 7], this iterative algorithm converges to a solution that satisfies all the constraints. The PEP algorithm can be seen as an adaptation of the adaboost algorithm to the setting of query release. Therefore, to solve (8), we use an iterative projection algorithm that on each round selects an unsatisfied constraint and moves the distribution by the smallest possible distance to satisfy it.

Let $\widetilde{Q}_{1:t}$ and $\widetilde{A}_{1:t}$ be the set of queries and noisy measurements obtained using the private selection mechanism. Let $K$ be the number of iterations during the optimization and let $D_{t,0}, \dots, D_{t,K}$ be the sequence of projections during the $K$ iteration of optimization. The goal is that $D_{t,K}$ matches all the constraints defined by $\widetilde{Q}_{1:t}, \widetilde{A}_{1:t}$. Our initial distribution is the uniform distribution $D_{t,0} = U$. Then

on round $k \in [K]$, the algorithm selects an index $i_k \in [t]$ such that the $i_k$-th constraint has high error on the current distribution $D_{t,k-1}$. Then the algorithm projects the distribution such that the $i_k$-th constraint is satisfied and the distance to $D_{t,k-1}$ is minimized. Thus, the objective for iteration $k$ is:

$$\text{minimize:} \quad \text{RE}\left(D \parallel D_{t,k-1}\right) \qquad \text{subject to:} \qquad \widetilde{a}_{i_k} = \widetilde{q}_{i_k}(D), \quad \sum_{x \in \mathcal{X}} D(x) = 1 \qquad (10)$$

---

**Algorithm 4:** Exponential Weights Projection

---

**Input:** Error tolerance $\gamma$, linear queries $\widetilde{Q}_{1:T} = \{\widetilde{q}_1, \ldots, \widetilde{q}_T\}$, and noisy measurements $\widetilde{A}_{1:T} = \{\widetilde{a}_1, \ldots, \widetilde{a}_T\}$.
**Objective:** Minimize $\text{RE}\left(D \parallel U\right)$ such that $\forall_{i \in [T]} \quad |\widetilde{q}_i(D) - \widetilde{a}_i| \leq \gamma$.
Initialize $D_0$ to be the uniform distribution over $\mathcal{X}$, and $t \leftarrow 0$.
**while** $\max_{i \in [T]} |\widetilde{a}_i - \widetilde{q}_i(D_t)| > \gamma$ **do**
    **Choose:** $i \in [T]$ with $i \leftarrow \arg\max_{j \in [T]} |\widetilde{a}_j - \widetilde{q}_j(D_t)|$.
    **Update:** For all $x \in \mathcal{X}$, set $D_{t+1}(x) \leftarrow D_t(x) e^{-\lambda_t \widetilde{q}_i(x)}$, where $-\lambda_t = \ln\left(\frac{\widetilde{a}_i(1 - \widetilde{q}_i(D_t))}{(1 - \widetilde{a}_i)\widetilde{q}_i(D_t)}\right)$.
    $t \leftarrow t + 1$
**end**
**Output:** $D_T$

---

Then the Lagrangian of objective (10) is:

$$\mathcal{L}(D, \lambda) = \text{RE}\left(D \parallel D_{k-1}\right) + \lambda\left(\widetilde{q}_{i_k}(D) - \widetilde{a}_{i_k}\right) + \mu\left(\sum_{x \in \mathcal{X}} D(x) - 1\right) \qquad (11)$$

Taking the partial derivative with respect to $D(x)$, we have

$$\frac{\partial \mathcal{L}(D, \lambda)}{\partial D(x)} = \ln\left(\frac{D(x)}{D_{k-1}(x)}\right) + 1 + \lambda \widetilde{q}_{i_k}(x) + \mu = 0 \qquad (12)$$

Solving (12) for $D(x)$, we get

$$D(x) = D_{k-1}(x) \exp\left(-\lambda \widetilde{q}_{i_k}(x) - 1 - \mu\right) = \frac{D_{k-1}(x)}{Z} \exp\left(-\lambda \widetilde{q}_{i_k}(x)\right) \qquad (13)$$

where $\mu$ is chosen to satisfy the constraint $\sum_{x \in \mathcal{X}} D(x) = 1$ and $Z$ is a regularization factor. Plugging (13) into (11), we get:

$$
\begin{aligned}
\mathcal{L}(D, \lambda) &= \text{RE}\left(D \parallel D_{k-1}\right) + \lambda\left(\widetilde{q}_{i_k}(D) - \widetilde{a}_{i_k}\right) \\
&= \sum_x D(x) \log\left(\frac{D(x)}{D_{k-1}(x)}\right) + \lambda\left(\widetilde{q}_{i_k}(D) - \widetilde{a}_{i_k}\right) \\
&= \sum_x D(x) \log\left(\frac{1}{Z}\frac{D_{k-1}(x)e^{-\lambda q_{i_k}(x)}}{D_{k-1}(x)}\right) + \lambda\left(\widetilde{q}_{i_k}(D) - \widetilde{a}_{i_k}\right) \qquad (13) \\
&= -\lambda \sum_x D(x)\widetilde{q}_{i_k}(x) - \log(Z) + \lambda\left(\widetilde{q}_{i_k}(D) - \widetilde{a}_{i_k}\right) \\
&= -\lambda \widetilde{q}_{i_k}(D) - \log(Z) + \lambda\left(\widetilde{q}_{i_k}(D) - \widetilde{a}_{i_k}\right) \\
&= -\log(Z) - \lambda \widetilde{a}_{i_k} \\
&= -\log\left(\sum_{x \in \mathcal{X}} D_{k-1}(x) \exp\left(-\lambda \widetilde{q}_{i_k}(x)\right)\right) - \lambda \widetilde{a}_{i_k}
\end{aligned}
$$

The next step is to find the optimal value of $\lambda$. Therefore we calculate the derivative of $\mathcal{L}(D, \lambda)$ with respect to $\lambda$:

$$
\begin{aligned}
\frac{\partial \mathcal{L}(D, \lambda)}{\partial \lambda} &= \frac{e^{-\lambda}\widetilde{q}_{i_k}(D_{k-1})}{\sum_{x \in \mathcal{X}} D_{k-1}(x) \exp\left(-\lambda \widetilde{q}_{i_k}(x)\right)} - \widetilde{a}_{i_k} \\
&= \frac{e^{-\lambda}\widetilde{q}_{i_k}(D_{k-1})}{e^{-\lambda}\widetilde{q}_{i_k}(D_{k-1}) + (1 - \widetilde{q}_{i_k}(D_k))} - \widetilde{a}_{i_k}
\end{aligned}
$$

Setting $\frac{\partial \mathcal{L}(D,\lambda)}{\partial \lambda} = 0$, we can solve for $\lambda$.

$$\frac{e^{-\lambda}\widetilde{q}_{i_k}(D_{k-1})}{e^{-\lambda}\widetilde{q}_{i_k}(D_{k-1}) + (1 - \widetilde{q}_{i_k}(D_k))} = \widetilde{a}_{i_k}$$

$$e^{-\lambda}\widetilde{q}_{i_k}(D_{k-1}) = \widetilde{a}_{i_k}\left(e^{-\lambda}\widetilde{q}_{i_k}(D_{k-1}) + (1 - \widetilde{q}_{i_k}(D_k))\right)$$

$$e^{-\lambda}\widetilde{q}_{i_k}(D_{k-1}) - e^{-\lambda}\widetilde{a}_{i_k}\widetilde{q}_{i_k}(D_{k-1}) = \widetilde{a}_{i_k}\left(1 - \widetilde{q}_{i_k}(D_k)\right)$$

$$e^{-\lambda}\widetilde{q}_{i_k}(D_{k-1})\left(1 - \widetilde{a}_{i_k}\right) = \widetilde{a}_{i_k}\left(1 - \widetilde{q}_{i_k}(D_k)\right)$$

Finally we obtain

$$-\lambda = \ln\left(\frac{\widetilde{a}_{i_k}\left(1 - \widetilde{q}_{i_k}(D_{k-1})\right)}{(1 - \widetilde{a}_{i_k})\widetilde{q}_{i_k}(D_{k-1})}\right)$$

# D  Additional empirical evaluation

## D.1  Experimental details

We present hyperparameters used for methods across all experiments in Tables 1, 2, 3, 4, and 5. To limit the runtime of PEP and PEP$^{\mathsf{Pub}}$, we add the hyperparameter, $T_{max}$, which controls the maximum number of update steps taken at each round $t$. Our implementations of MWEM, DualQuery, and PMW$^{\mathsf{Pub}}$ are adapted from `https://github.com/terranceliu/pmw-pub`. We implement RAP and RAP$^{\mathsf{softmax}}$ ourselves using PyTorch since the code for RAP. All experiments are run using a desktop computer with an Intel® Core™ i5-4690K processor and NVIDIA GeForce GTX 1080 Ti graphics card.

We obtain the ADULT and ACS datasets by following the instructions outlined in `https://github.com/terranceliu/pmw-pub`. Our version of ADULT used to train GEM$^{\mathsf{Pub}}$ (reduced) (Figure 2b and 7) uses the following attributes: sex, race, relationship, marital-status, occupation, education-num, age.

Table 1: PEP hyperparameters

| Dataset | Parameter | Values |
|---|---|---|
| All | $T_{max}$ | 25 |
| ACS (red.) | $T$ | 20, 30, 40, 50, 75 100, 125, 150, 175, 200 |
| ADULT (red.) | $T$ | 20, 30, 40, 50, 75 100, 125, 150, 175, 200 |

Table 2: GEM hyperparameters

| Dataset | Parameter | Values |
|---|---|---|
| All | hidden layer sizes | $(512, 1024, 1024)$ |
| | learning rate | 0.0001 |
| | $B$ | 1000 |
| | $\alpha$ | 0.67 |
| | $T_{max}$ | 100 |
| ACS | $T$ | 100, 150, 200, 250, 300, 400, 500, 750, 1000 |
| ACS (red.) | $T$ | 50, 75, 100, 125, 150, 200, 250, 300 |
| ADULT, ADULT (red.), ADULT (orig), LOANS | $T$ | 30, 40, 50, 60, 70, 80, 90, 100, 125, 150, 175, 200 |

Table 3: PEP$^{\mathsf{Pub}}$ hyperparameters

| Dataset | Parameter | Values |
|---|---|---|
| All | $T_{max}$ | 25 |
| ACS | $T$ | 20, 40, 60, 80, 100 120, 140, 160, 180, 200 |

## D.2  Main experiments with additional metrics

In Figures 5, 6, and 7, we present the same results for the same experiments described in Section 7.1 (Figures 1 and 2), adding plots for mean error and root mean squared error (RMSE). For our

Table 4: GEMPub hyperparameters

| Dataset | Parameter | Values |
|---|---|---|
| All | hidden layer sizes | $(512, 1024, 1024)$ |
|  | learning rate | 0.0001 |
|  | $B$ | 1000 |
|  | $\alpha$ | 0.67 |
|  | $T_{max}$ | 100 |
| ACS | $T$ | 30, 40, 50, 75, 100, 150, 200, 300, 400, 500 |
| ADULT | $T$ | 2, 3, 5, 10, 20, 30, 40, 50, 60 70 80 90 100 |

Table 5: Baseline hyperparameters

| Method | Parameter | Values |
|---|---|---|
| RAP | learning rate | 0.001 |
|  | $n'$ | 1000 |
|  | $K$ | 5, 10, 25, 50, 100 |
|  | $T$ | 2, 5, 10, 25, 50, 75, 100 |
| RAPsoftmax | learning rate | 0.1 |
|  | $n'$ | 1000 |
|  | $K$ | 5, 10, 25, 50, 100 |
|  | $T$ | 2, 5, 10, 25, 50, 75, 100 |
| MWEM | $T$ | 100, 150, 200, 250, 300 400, 500, 750, 1000 |
| MWEM (/w past queries) | $T$ | 50, 75, 100, 150, 200, 250, 300 |
| DualQuery | $\eta$ | 2, 3, 4, 5 |
|  | samples | 25 50, 100, 250, 500 |

experiments on ACS PA-18 with public data, we add results using 2018 data for Ohio (ACS OH-18), which we note also low *best-mixture-error*. Generally, the relative performance between the methods for these other two metrics is the same as for max error.

In addition, in Figure 6, we present results for PEPPub, a version of PEP similar to PMWPub that is adapted to leverage public data (and consequently can be applied to high dimensional settings). We briefly describe the details below.

PEPPub. Like in Liu et al. [26], we extend PEP by making two changes: (1) we maintain a distribution over the public data domain and (2) we initialize the approximating distribution to that of the public dataset. Therefore like PMWPub, PEPPub also restricts $\mathcal{D}$ to distributions over the public data domain and initializes $D_0$ to be the public data distribution.

We note that PEPPub performs similarly to PMWPub, making it unable to perform well when using ACS CA-18 as a public dataset (for experiments on ACS PA-18). Similarly, it cannot be feasibly run for the ADULT dataset when the public dataset is missing a significant number of attributes.

### D.3 Comparisons against RAP

In Figure 8, we show failures cases for RAP. Again, we see that RAPsoftmax outperforms RAP in every setting. However, we observe that aside from ADULT (reduced), RAP performs extremely poorly across all privacy budgets.

To account for this observation, we hypothesize that by projecting each measurement to Aydore et al. [5]'s proposed continuous relaxation of the synthetic dataset domain, RAP produces a synthetic

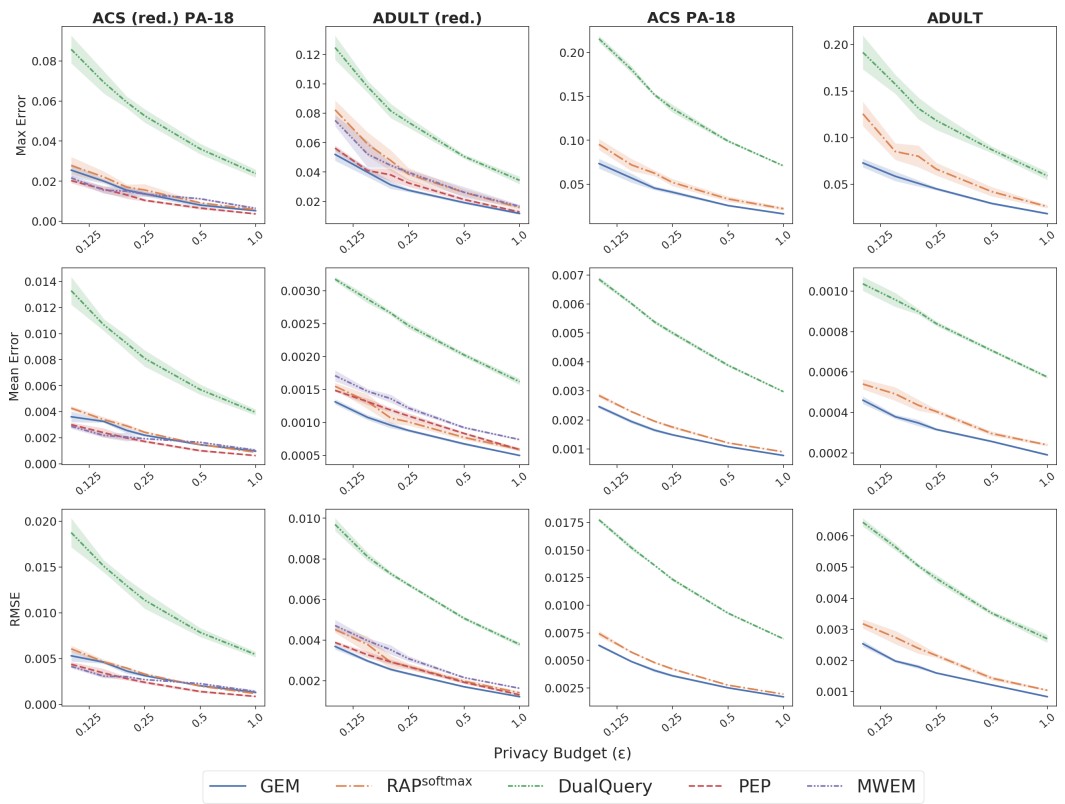

Figure 5: Max, mean, and root mean squared errors for 3-way marginals evaluated on ADULT and ACS PA-18 using privacy budgets $\varepsilon \in \{0.1, 0.15, 0.2, 0.25, 0.5, 1\}$ and $\delta = \frac{1}{n^2}$. The *x-axis* uses a logarithmic scale. We evaluate using the following workload sizes: ACS (reduced) PA-18: $455$; ADULT (reduced): $35$; ACS PA-18: $4096$; ADULT: $286$. Results are averaged over $5$ runs, and error bars represent one standard error.

dataset that is inconsistent with the semantics of an actual dataset. Such inconsistencies make it more difficult for the algorithm to do well without seeing the majority of high error queries.

Consider this simple example comparing GEM and RAP. Suppose we have some binary attribute $A \in \{0, 1\}$ and we have $P(A = 0) = 0.2$ and $P(A = 1) = 0.8$. For simplicity, suppose that the initial answers at $t = 0$ for both algorithms is 0 for the queries $q_{A=0}$ and $q_{A=1}$. Assume at $t = 1$ that the privacy budget is large enough such that both algorithms select the max error query $q_{A=1}$ (error of $0.8$), which gives us an error or $0.8$. After a single iteration, both algorithms can reduce the error of this query to $0$. In RAP, the max error then is $0.2$ (for the next largest error query $q_{A=0}$). However for GEM to output the correct answer for $q_{A=1}$, it must learn a distribution (due to the softmax activation function) such that $P(A = 1) = 0.8$, which naturally forces $P(A = 0) = 0.2$. In this way, GEM can reduce the errors of both queries in one step, giving it an advantage over RAP.

In general, algorithms within the Adaptive Measurements framework have this advantage in that the answers it provides must be consistent with the data domain. For example, if again we consider the two queries for attribute $A$, a simple method like the Gaussian or Laplace mechanism has a nonzero probability of outputting noisy answers for $q_{A=0}$ and $q_{A=1}$ such that $P(A = 0) + P(A = 1) \neq 1$. This outcome however will never occur in Adaptive Measurements.

Therefore, we hypothesize that RAP tends to do poorly as you increase the number of high error queries because the algorithm needs to select each high error query to obtain low error. Synthetic data generation algorithms can more efficiently make use of selected query measurements because their answers to all possible queries must be consistent. Referring to the above example again, there

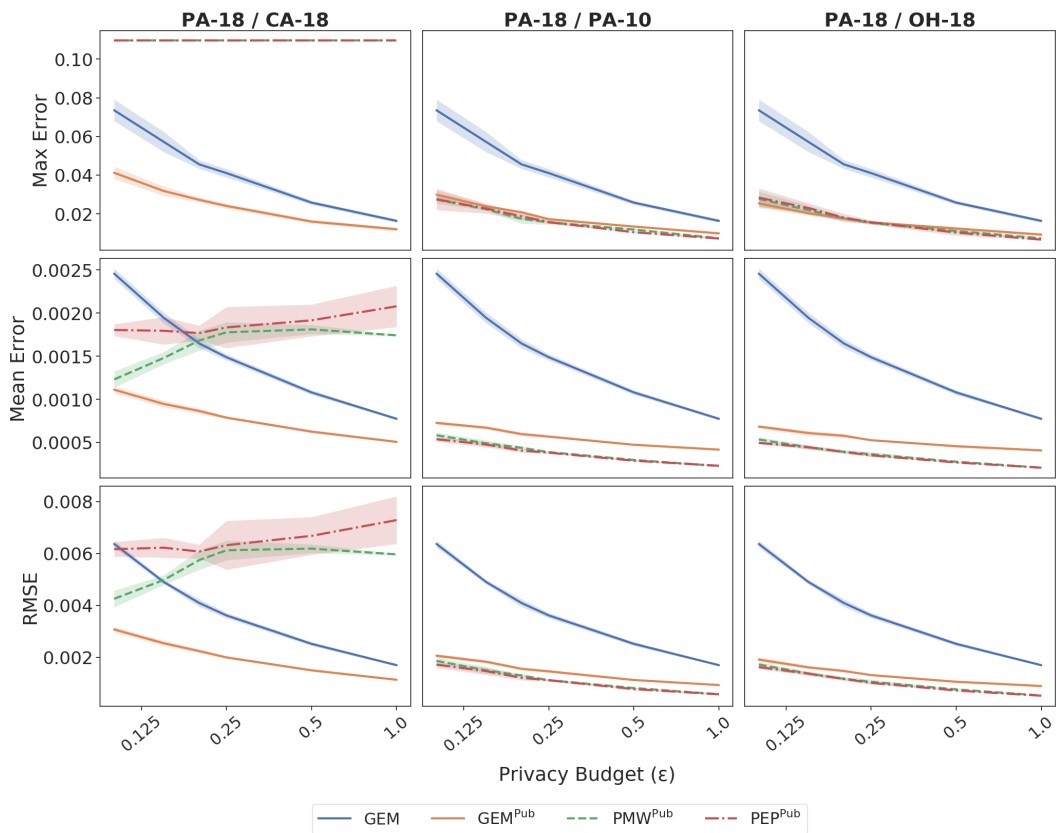

Figure 6: Max, mean, and mean squared error for 3-way marginals on ACS PA-18 (workloads = 4096) with privacy budgets $\varepsilon \in \{0.1, 0.15, 0.2, 0.25, 0.5, 1\}$ and $\delta = \frac{1}{n^2}$. We evaluate public-data-assisted algorithms with the following public datasets: **Left:** 2018 California (CA-18); **Center:** 2010 Pennsylvania (PA-10); **Right:** 2018 Ohio (PA-10). The *x-axis* uses a logarithmic scale. Results are averaged over 5 runs, and error bars represent one standard error.

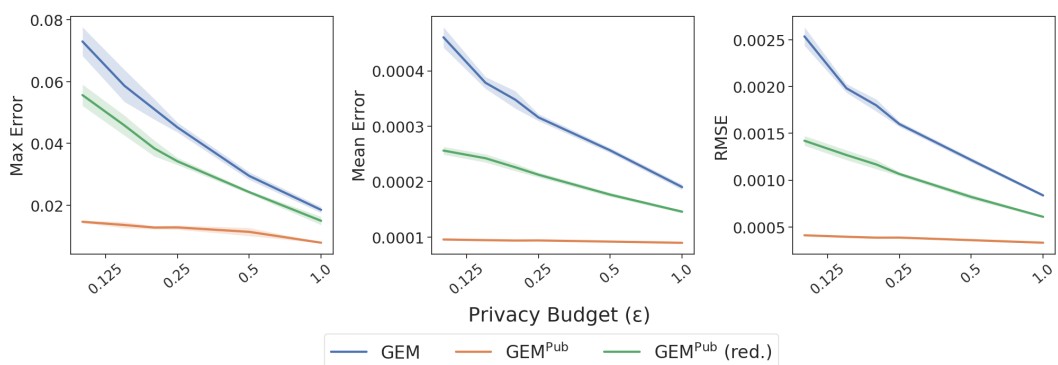

Figure 7: Max, mean, and mean squared error for 3-way marginals on ADULT (workloads = 286) with privacy budgets $\varepsilon \in \{0.1, 0.15, 0.2, 0.25, 0.5, 1\}$ and $\delta = \frac{1}{n^2}$. We evaluate GEM using both the complete public data (GEM$^{\mathsf{Pub}}$) and a reduced version that has fewer attributes (GEM$^{\mathsf{Pub}}$ (reduced)). The *x-axis* uses a logarithmic scale. Results are averaged over 5 runs, and error bars represent one standard error.

may exist two high error queries $q_{A=0}$ and $q_{A=1}$, but only one needs to be sampled to reduce the errors of both.

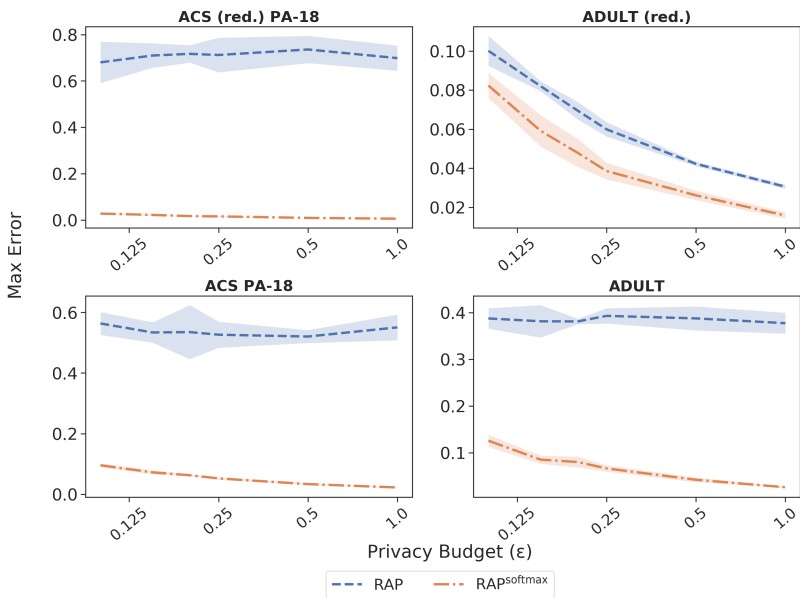

Figure 8: Comparison of RAP and RAP^softmax w.r.t max error for 3-way marginals evaluated on ADULT and ACS PA-18 using privacy budgets $\varepsilon \in \{0.1, 0.15, 0.2, 0.25, 0.5, 1\}$ and $\delta = \frac{1}{n^2}$. The *x-axis* uses a logarithmic scale. We evaluate using the following workload sizes: ACS (reduced) PA-18: 455; ACS PA-18: 4096; ADULT (reduced): 286; ADULT: 35. Results are averaged over 5 runs, and error bars represent one standard error.

We refer readers to Appendix D.7, where we use the above discussion to account for how the way in which the continuous attributes in ADULT are preprocessed can impact the effectiveness of RAP.

## D.4 Marginal trick

While this work follows the literature in which methods iteratively sample sensitivity 1 queries, we note that the marginal trick approach (Appendix A.1) can be applied to all iterative algorithms under Adaptive Measurements. To demonstrate this marginal trick's effectiveness, we show in Figure 9 how the performance of GEM improves across max, mean, and root mean squared error by replacing the *Update* and *Measure* steps in Adaptive Measurements.

In this experiment, given that the number of measurements taken is far greater when using the marginal trick, we increased $T_{max}$ for GEM from 100 to 10000 and changed the loss function from $\ell_1$-loss to $\ell_2$-loss. Additional hyperparameters used can be found in Table 6. Note that we reduced the model size for $G$ simply to speed up runtime. Overall, we admit that leveraging this trick was not our focus, and so we leave designing GEM (and other iterative methods) to fully take advantage of the marginal trick to future work.

## D.5 Discussion of HDMM

HDMM [28] is an algorithm designed to directly answer a set of workloads, rather than some arbitrary set of queries. In particular, HDMM optimizes some strategy matrix to represent each workload of queries that in theory, facilitates an accurate reconstruction of the workload answers while decreasing the sensitivity of the privacy mechanisms itself. In their experiments, McKenna et al. [28] show strong results w.r.t. RMSE, and the U.S. Census Bureau itself has incorporated aspects of the algorithm into its own releases [24].

We originally planned to run HDMM as a baseline for our algorithms in the standard setting, but after discussing with the original authors, we learned that currently, the available code for HDMM makes running the algorithm difficult for the ACS and ADULT datasets. There is no way to solve the least square problem described in the paper for domain sizes larger than $10^9$, and while the authors admit

Table 6: GEM (marginal trick) hyperparameters

| Dataset | Parameter | Values |
|---------|-----------|--------|
| All | hidden layer sizes | $(256, 512)$ |
|  | learning rate | $0.0001$ |
|  | $B$ | $1000$ |
|  | $\alpha$ | $0.5$ |
|  | $T_{max}$ | $10000$ |
| ACS | $T$ | 50, 100, 150, 200, 250, 300, 400, 500 |
| ACS (red.) | $T$ | 50, 100, 150, 200, 250, 300, 450 |
| ADULT | $T$ | 30, 40, 50, 75, 100, 125, 150, 200 |
| ADULT (red.) | $T$ | 5, 10, 15, 20 25, 30, 35 |

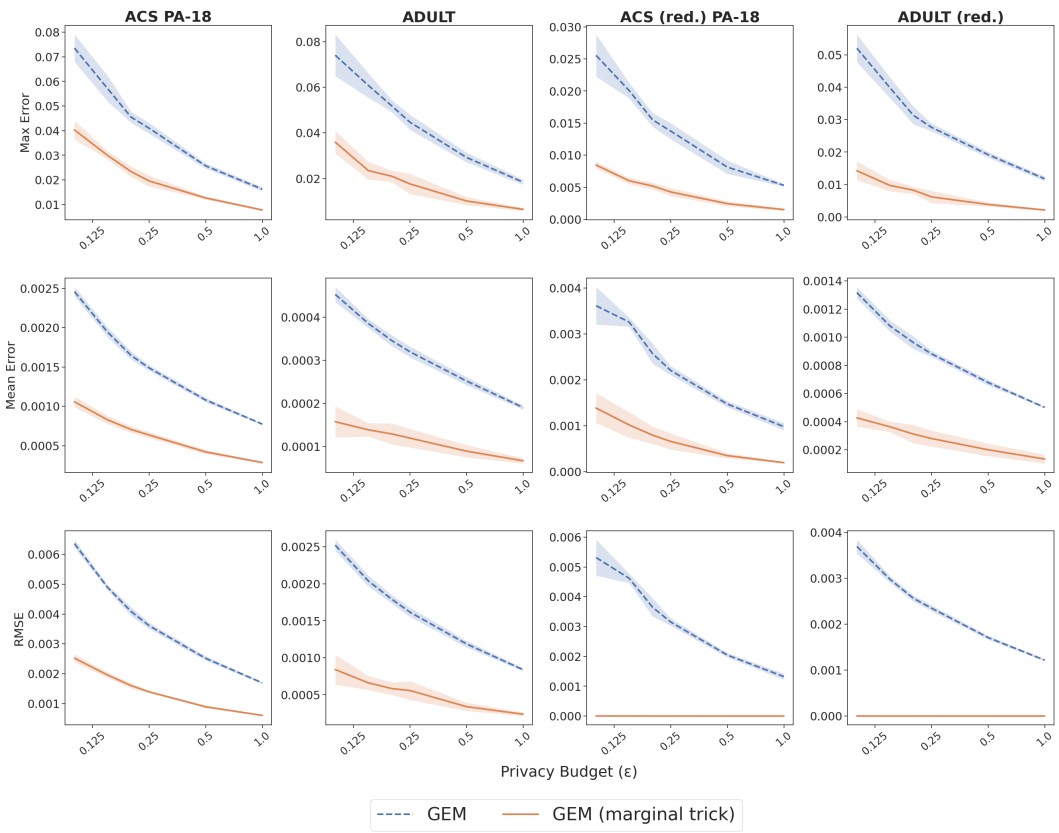

Figure 9: Error comparison of GEM with and without the marginal trick , evaluated on 3-way marginals with privacy budgets $\varepsilon \in \{0.1, 0.15, 0.2, 0.25, 0.5, 1\}$ and $\delta = \frac{1}{n^2}$. The *x-axis* uses a logarithmic scale. Results are averaged over 5 runs, and error bars represent one standard error.

that HDMM could possibly be modified to use local least squares for general workloads (outside of those defined in their codebase), this work is not expected to be completed in the near future.

We also considered running HDMM+PGM [29], which replaces the least squares estimation problem a graphical model estimation algorithm. Specifically, using (differentially private) measurements to some set of input queries, HDMM+PGM infers answers for any workload of queries. However, the

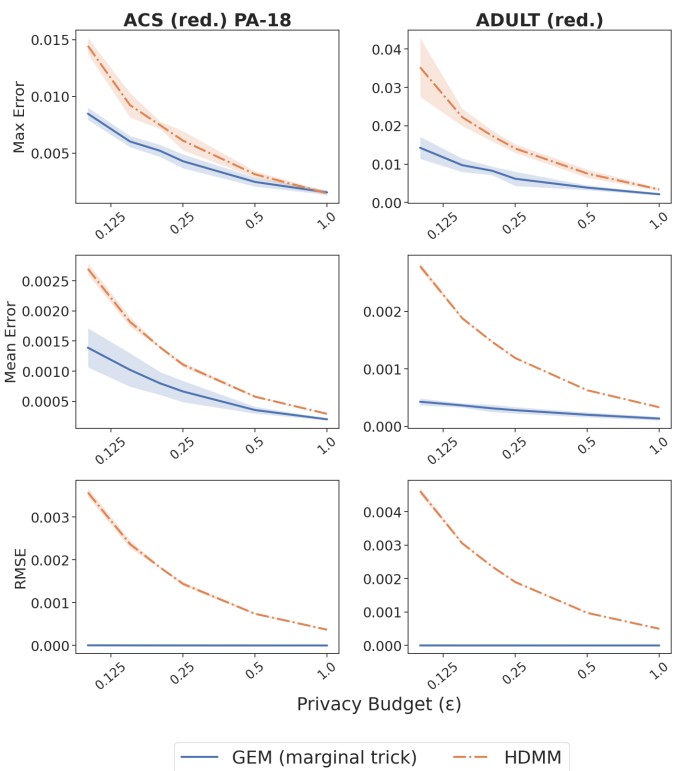

Figure 10: Comparison of max, mean, and root mean squared errors against HDMM on ACS (reduced) PA-18 (workloads=455) and ADULT (reduced) (workloads=35), evaluated on 3-way marginals with privacy budgets $\varepsilon \in \{0.1, 0.15, 0.2, 0.25, 0.5, 1\}$ and $\delta = \frac{1}{n^2}$. The *x-axis* uses a logarithmic scale. Results are averaged over 5 runs, and error bars represent one standard error.

memory requirements of the algorithm scale exponentially with dimension of the maximal clique of the measurements, prompting users to carefully select measurements that help build a useful junction tree that is not too dense. Therefore, the choice of measurements and cliques can be seen as hyperparameters for HDMM+PGM, but as the authors pointed out to us, how such measurements should be selected is an open problem that hasn't been solved yet. In general, cliques should be selected to capture correlated attributes without making the size of the graphical model intractable. However, we were unsuccessful in finding a set of measurements that achieved sensible results (possibly due to the large number of workloads our experiments are designed to answer) and decided stop pursuing this endeavor due to the heavy computational resources required to run HDMM+PGM. We leave finding a proper set of measurements for ADULT and ACS PA-18 as an open problem.

Given such limitations, we evaluate HDMM with least squares on ACS (reduced) PA-18 and ADULT (reduced) only (Figure 10). We use the implementation found in `https://github.com/ryan112358/private-pgm`. We compare to GEM using the marginal trick, which HDMM also utilizes by default. While GEM outperforms HDMM, HDMM seems to be very competitive on low dimensional datasets when the privacy budget is higher. In particular, HDMM slightly outperforms GEM w.r.t. max error on ACS (reduced) when $\varepsilon = 1$. We leave further investigation of HDMM and HDMM+PGM to future work.

### D.6 Effectiveness of optimizing over past queries

One important part of the adaptive framework is that it encompasses algorithms whose update step uses measurements from past iterations. In Figure 11, we verify claims from Hardt et al. [22] and Liu et al. [26] that empirically, we can significantly improve over the performance of MWEM when incorporating past measurements.

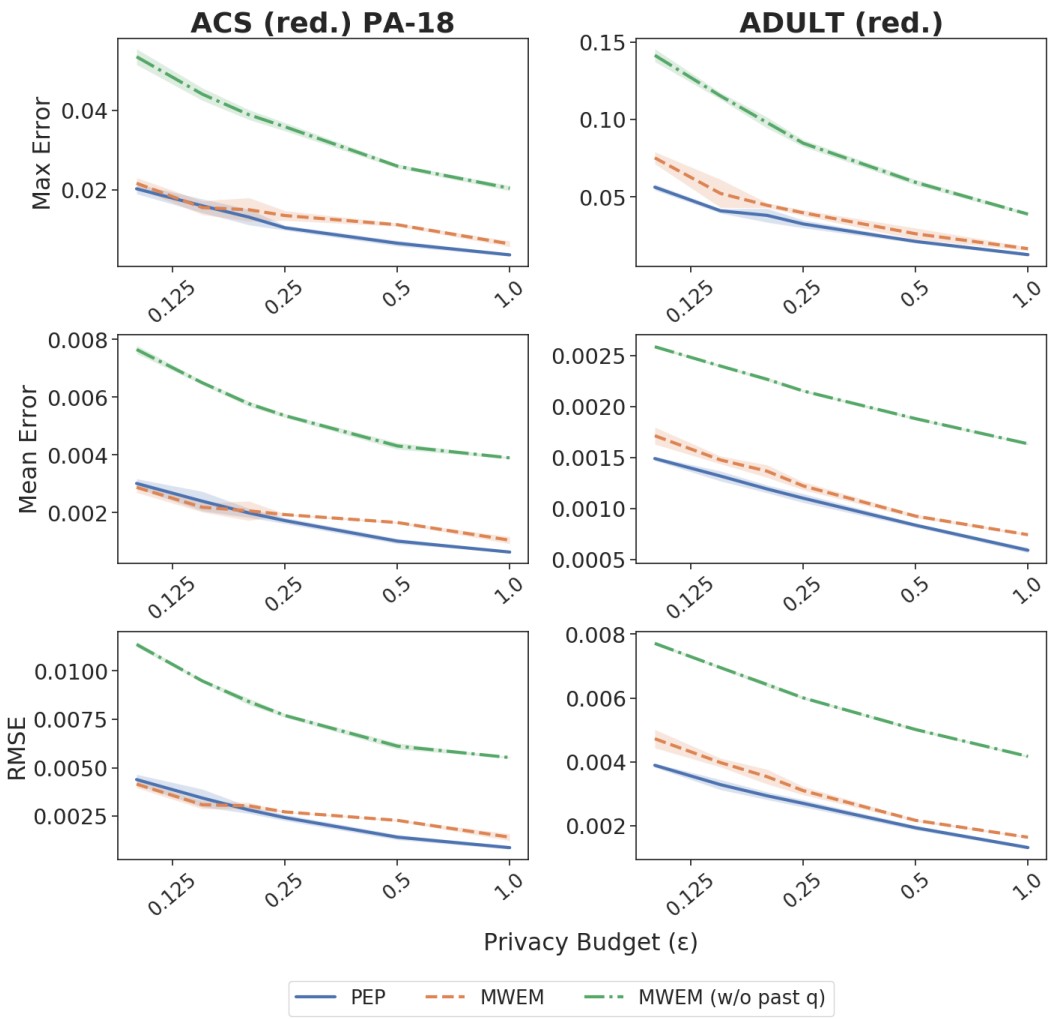

Figure 11: Comparison of max, mean, and root mean squared errors against vanilla MWEM that does not use queries sampled during past iterations, evaluated on 3-way marginals with privacy budgets $\varepsilon \in \{0.1, 0.15, 0.2, 0.25, 0.5, 1\}$ and $\delta = \frac{1}{n^2}$. The *x-axis* uses a logarithmic scale. Results are averaged over 5 runs, and error bars represent one standard error.

### D.7 Evaluating on ADULT* and LOANS

In Figure 12, we reproduce the experiments on the ADULT (which we denote as ADULT*) and LOANS datasets presented in Aydore et al. [5]. Like Aydore et al. [5], we obtain the datasets from https://github.com/giusevtr/fem. Empirically, we find that GEM outperforms all baseline methods. In addition, while RAP performs reasonably well, we observe that by confining $\mathcal{D}$ to $\left\{\sigma(M)|M \in \mathbb{R}^{n' \times d}\right\}$ with the softmax function, RAP$^{\text{softmax}}$ performs better across all privacy budgets.

To account for why RAP performs reasonably well with respect to max error on ADULT* and LOANs but very poorly on ADULT and ACS, we refer back to our discussion about the issues of RAP presented in Appendix D.3 in which argue that by outputting synthetic data that is inconsistent with any real dataset, RAP performs poorly when there are many higher error queries. ADULT* and LOANs are preprocessed in a way such that continous attributes are converted into categorical (technically ordinal) attributes, where a separate categorical value is created for each unique value that the continuous attribute takes on in the dataset (up to a maximum of 100 unique values). When processed in this way, $k$-way marginal query answers are sparser, even when $k$ is relatively small

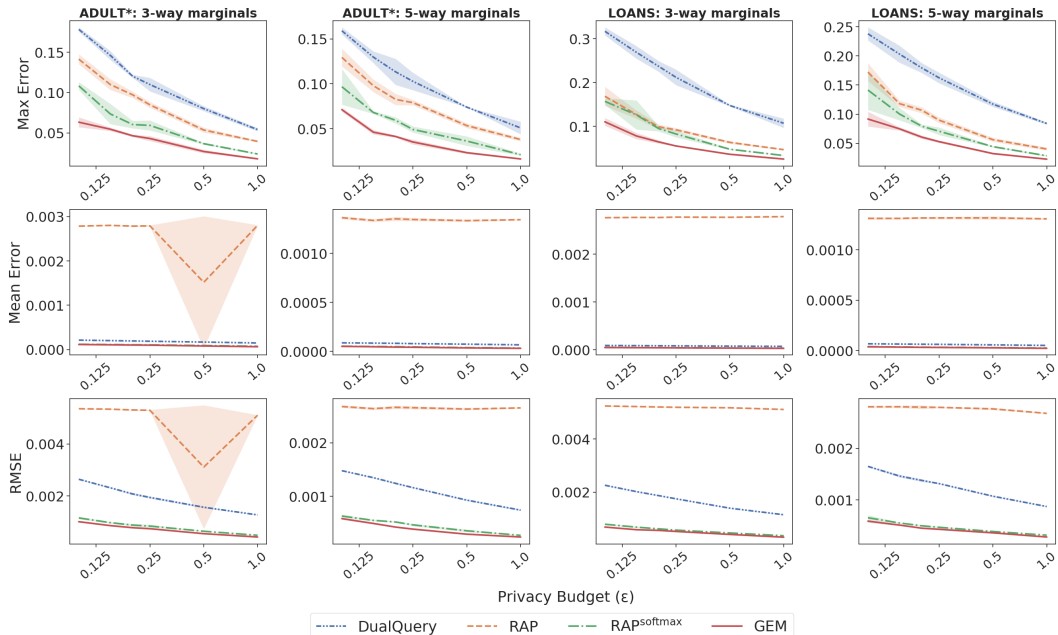

Figure 12: Max, mean, and root mean squared errors for 3-way marginals with a workload size of $64$. Methods are evaluated on ADULT* and LOANS datasets using privacy budgets $\varepsilon \in \{0.1, 0.15, 0.2, 0.25, 0.5, 1\}$ and $\delta = \frac{1}{n^2}$. The *x-axis* uses a logarithmic scale. Results are averaged over $5$ runs, and error bars represent one standard error.

($\leq 5$). However, Liu et al. [26] preprocess continuous variables in the ADULT and ACS dataset by constructing bins, resulting in higher error queries.

For example, suppose in an unprocessed dataset (with $n$ rows), you have 3 rows where an attribute (such as income) takes on the values $16,587, 15,984$, and $18,200$. Next, suppose there exists datasets A and B, where dataset A maps each unique value to its own category, while dataset B constructs a bin for values between $15,000$ and $20,000$. Then considering all 1-way marginal queries involving this attribute, dataset A would have 3 different queries, each with answer $\frac{1}{N}$. Dataset B however would only have a single query whose answer is $\frac{3}{N}$. Whether a dataset should be preprocessed as dataset A or dataset B depends on the problem setting.[6] However, this (somewhat contrived) example demonstrates how dataset B would have more queries with high value answers (and therefore more queries with high initial errors, assuming that the algorithms in question initially outputs answers that are uniform/close to $0$).

In our experiments with 3-way marginal queries, ADULT (where workload is $286$) and ADULT* (where the workload is $64$) have roughly the same number queries ($334, 128$ vs. $458, 996$ respectively). However, ADULT has $487$ queries with answers above $0.1$ while ADULT* only has $71$. Looking up the number of queries with answers above $0.2$, we count $181$ for ADULT and only $28$ for ADULT*. Therefore, experiments on ADULT* have fewer queries that RAP needs to optimize over to achieve low max error, which we argue accounts for the differences in performance on the two datasets.

Finally, we note that in Figure 8, RAP has relatively high mean error and RMSE. We hypothesize that again, because only the queries selected on each round are optimized and all other query answers need not be consistent with the optimized ones, RAP will not perform well on any metric that is evaluated over all queries (since due to privacy budget constraints, most queries/measurements are never seen by the algorithm). We leave further investigation on how RAP operates in different settings to future work.

---

[6]We would argue that in many cases, dataset B makes more sense since it is more likely for someone to ask—"How many people make between $15,000$ and $20,000$ dollars?"—rather than—"How many people make $15,984$ dollars?".