# OpenReview forum: "Iterative Methods for Private Synthetic Data: Unifying Framework and New Methods"
_NeurIPS.cc/2021/Conference — NeurIPS 2021 Poster_

### Official Review · Reviewer_ZboG · 2021-07-09

**Rating:** 6
**Confidence:** 3

**Summary:**

# Summary of contributions

The paper studies the problem of synthesizing differentially private (DP) data for query release. Here we are given a sensitive dataset $P$ together with a set $\mathcal{Q}$ of queries of interest. The goal is to generate a DP synthetic dataset $D$ so that answering the queries using $D$ gives similar answers to those of $P$; more formally, we want to minimize the maximum error across the queries, i.e. $\max_{q \in \mathcal{Q}} |q(P) - q(D)|$. Several works have proposed algorithms for this problem, including MWEM [Hardt et al., NeurIPS'12] and FEM [Vietri et al., ICML'20].

The first contribution of the paper is in observing that essentially the previously proposed algorithms all fit into a single framework which they call "Adaptive Measurements". Specifically, the algorithm proceeds in iterations and, in each iteration, it uses the exponential mechanism to select one or more queries, asks for noisy values of those queries and it then updates the current synthetic dataset $D$ by minimizing some objective (which involves the answers to the queries so far and the current guess dataset).

With the above in mind, the authors then decide a new algorithm which fits into this framework called Private Entropy Projection (PEP) algorithm. At each step, PEP tries to find the distribution $D$ with maximum entropy among those whose max error w.r.t. the noisy measurements is at most $\gamma$ for some parameter $\gamma$. The authors note that PEP also gives exponentially weighted distributions, similar to MWEM, but the weights are more elaborated. Empirically, the authors show that PEP performs better than MWEM when $\epsilon$ is sufficiently large (and performs similarly for small $\epsilon$) on reduced versions of ADULT and ACS datasets for 3-way marginal queries.

The reason a reduced version of those datasets are used is because the original datasets have high-dimension, which is infeasible for MWEM or PEP since they have to keep the entire distribution explicitly. To overcome this, the authors propose a different algorithm called GEM which instead assume that the distribution is generated to some network $G_\theta$ with parameter $\theta$ (assuming that the inputs to the network is i.i.d. Gaussian). This can help reduce the computational cost dramatically since now only the network parameters $\theta$ is needed to be maintained. Here, GEM still works in the "Adaptive Measurements" framework and the authors use gradient descent to minimize the $\ell_1$ loss w.r.t. the noisy measurements so far. Note that this might not be easily differentiable for generic queries; fortunately, for k-way marginal queries, the authors point out that they are just product of the network's outputs and as a result the gradient can be easily found.

Finally, the authors consider the setting where there is some public data which one hopes to use to improve the quality of the synthetic dataset. Previous state-of-the-art called PMW$^{Pub}$ [Liu et al., 2021] simply uses limits the distribution to only those on the _support_ of the public data and runs MWEM. This has several limitations, such as when the support of the public data is not sufficient to express the sensitive data, or when some features are missing in the public dataset. The authors show that a modified version of GEM (called GEM$^{Pub}$) is more amenable for these cases: just use the public data to find an initial generator network, and then proceed with the GEM algorithm as usual. They demonstrate that this approach indeed works well in practice: although the performances of GEM$^{Pub}$ and PMW$^{Pub}$ are similar for large & similarly-distributed public data, GEM$^{Pub}$ significantly outperforms PMW$^{Pub}$ when public data is incomplete or its support not closely resembling the sensitive data.

**Limitations And Societal Impact:**

I do not see any significant negative social impact of this work. One thing that might be useful is to discuss the issue of bias that might arise in the public-assisted setting.

**Main Review:**

# Evaluation

## Strengths

- The GEM algorithm is surprisingly intuitive and also works very well in the public-assisted setting. This can potentially be helpful as many recent research works in DP have been trying to exploit public data, but as far as I am aware of, none of the proposed solutions is as robust as the new GEM$^{Pub}$ algorithm.

## Weaknesses

- My main criticism is in regards to the authors' descriptions of their algorithms. There are often confusing, imprecise and sometimes inconsistent. Please see below when I go into more detail on this issue for the PEP algorithm. (A summary is that the algorithm described in the main body is not exactly the same as that described in the appendix which itself is also slightly different from what was actually implemented.)
- There is no theoretical utility guarantee for PEP. This makes it much harder to compare to previous algorithms like MWEM, FEM or DualQuery, all of which have good formal guarantees on the maximum error. (Note that the lack of theoretical utility guarantee for GEM is more understandable since it might require complicated assumption on the generator network, but the worst-case guarantee for PEP should exist without any additional assumption.)
- There are also quite a fair bit of typos & errors in writing. For example, their privacy proof completely forgets to take $k$ (number of queries selected per stage) into account altogether.

## Recommendation

Overall, although I think that the paper should be significantly revised (so that the algorithm descriptions are more clear & consistent) and that theoretical guarantees should be derived for PEP, I still think the new public-assisted approach can be useful both to the research community and in practice.

# Detailed Comments

- **Issues with PEP Algorithm**:
  - First of all, in the main body the selection of $D$ is described as maximizing the entropy subject to the constraints $|\tilde{a}_i - \tilde{q}_i(D)| \leq \gamma$. What happens if the constraints cannot be satisfied (i.e. when $\tilde{a}_i$ has a lot of noise)? Also, $\gamma$ is never specified anywhere; how is it selected?
  - Secondly, the algorithm you are using (Algorithm 2) is only trying to fix one of these constraints at a time. I don't see why this should even converge, let alone have a good convergence rate.
  - Finally, your code (`project` function in `pep.py`) does not even take $\gamma$ at all. It just tries to perform correct for the worst $\tilde{q}_i$ for a given number of iterations. It is unclear to me what kind of objective this is even trying to optimize.
- **Issue with Privacy Analysis**: The noise setting in Section 3.1 and proof of Theorem 1 in Appendix A do not take $k$ (number of queries selected per iteration) into account. Please fix this.

Minor comments:
- **Indices in Algorithm 1 vs in other places**: Please check the indices of $\tilde{q}, \tilde{a}$ and make sure they are all consistent. For example, the indices in Algorithm 1 here are $t,i$, but in MWEM (line 135-136), the index is $i$ and the sum is over $i = 1$ to $t$. These are pretty confusing; I think $i$ should be changed to $t$ here and $t$ to $T$. Similar inconsistencies can be found elsewhere too.

**Time Spent Reviewing:**

4.5

---

> ### Author Response · Authors · 2021-08-10
> **Response to Reviewer ZboG**
>
> Thank you for your detailed comments and suggestions.
>
> Regarding your minor comments, we will make sure to correct any inconsistencies in our notation. We answer your detailed comments below.
>
> **Issues with PEP Algorithm**
>
> ***Q:*** *What happens if the constraints cannot be satisfied (i.e. when $\tilde{a}_i$ has a lot of noise)? Also, $\gamma$ is never specified anywhere; how is it selected?*
>
> In principle, $\gamma$ can be chosen to be a high-probability upper bound on the noise, which can be calculated through standard concentration bounds on Gaussian noise. In that case, every constraint $|q(D) - \tilde{a}| \leq \gamma$ can be satisified and the optimization problem is well defined. However, we note that our algorithm is well defined for every choice of $\gamma \geq 0$. For example, in our experiments we have $\gamma=0$, and we obtain good empirical results that outperform MWEM (see response to question 3 regarding pep.py).
>
> ***Q:*** *Secondly, the algorithm you are using (Algorithm 2) is only trying to fix one of these constraints at a time.*
>
> To explain how algorithm 2 converges, we cite an established convergence analysis of adaboost from [Schapire, Robert E., and Yoav Freund, 2013, chapter 7]. Similar to adaboost, Algorithm 2 is running iterative projection, where on each iteration, it projects the distribution to satisfy a single constraint. As shown in [Schapire, Robert E., and Yoav Freund, 2013], this iterative algorithm converges to a solution that satisfies all the constraints. The PEP algorithm can be seen as an adaptation of the adaboost algorithm to the setting of query release. We will include this discussion (and relevant citations) in our revision.
>
> ***Q:*** *Finally, your code (project function in pep.py) does not even take $\gamma$ at all. It just tries to perform correct for the worst $q_i$ for a given number of iterations. It is unclear to me what kind of objective this is even trying to optimize.*
>
> One can think of $\gamma$ as a hyperparameter to the algorithm. However, in our code we simply set $\gamma = 0$ and (as you pointed out) add an additional parameter for the number of iterations, which allows us to control the runtime of our algorithm. Having added this modification, we show that PEP can still outperform MWEM without us having to fine tune $\gamma$. We will make this point clear in the Experimental Details of Section 7 (Empirical Evaluation), as well as in the algorithm details for PEP presented in the appendix.
>
> **Issue with Privacy Analysis**
>
> We will fix the proof in our revision. Thank you for pointing this out to us.

---

> > ### Comment · Reviewer_ZboG · 2021-08-17
> > **Re Response**
> >
> > Thanks the authors for the response. I'd encourage incorporating these explanations into the revision.
> >
> > Regarding the comment "we show that PEP can still outperform MWEM without us having to fine tune $\gamma$", do you know how much the choice of the number of iterations affect the empirical performance of PEP? If so, it might be good to discuss that in Section 7 too as this is now a parameter of the algorithm.

---

> > > ### Author Response · Authors · 2021-08-20
> > > **PEP iterations parameter**
> > >
> > > Thank you for the helpful comments.
> > >
> > > Regarding the number of iterations or projection steps: Running with more iterations (max_iter) does not hurt the algorithm's accuracy. Therefore, in principle, we can apply the projection step until the error on the previous measurements stops decreasing, but we cap the number of iterations to control the algorithms' runtime. In general, there is an accuracy/runtime trade-off with the number of projection steps. However, we chose a value (max_iter=25) that gives us reasonable runtime and still allows us to outperform MWEM. Here is a table showing how the performance changes for a fixed $\varepsilon=0.1$ and $T=30$ when we vary the number of projection steps. We will follow your recommendation and add this discussion to our paper.
> > >
> > > The tables bellow show that after about 5 projection steps, having more iterations does not improve the accuracy by much.
> > >
> > > We include results for two datasets:
> > >
> > > Results for the Adult (reduced) dataset:
> > >
> > > | max_iter 	| mean max-error 	| std max-error 	|
> > > |---	|---	|---	|
> > > | 1 	| 0.058 	| 0.012 	|
> > > | 2 	| 0.047 	| 0.004 	|
> > > | 3 	| 0.045 	| 0.004 	|
> > > | 4 	| 0.038 	| 0.002 	|
> > > | 5 	| 0.041 	| 0.009 	|
> > > | 6 	| 0.041 	| 0.005 	|
> > > | 7 	| 0.039 	| 0.004 	|
> > > | 8 	| 0.036 	| 0.005 	|
> > > | 9 	| 0.037 	| 0.005 	|
> > > | 10 	| 0.039 	| 0.004 	|
> > >
> > > Results for ACS (reduced) PA-18 dataset
> > >
> > > | max_iter 	| mean max-error 	| std max-error 	|
> > > |---	|---	|---	|
> > > | 1 	| 0.045 	| 0.003 	|
> > > | 2 	| 0.030 	| 0.004 	|
> > > | 3 	| 0.029 	| 0.005 	|
> > > | 4 	| 0.023 	| 0.006 	|
> > > | 5 	| 0.023 	| 0.003 	|
> > > | 6 	| 0.020 	| 0.004 	|
> > > | 7 	| 0.021 	| 0.001 	|
> > > | 8 	| 0.019 	| 0.002 	|
> > > | 9 	| 0.022 	| 0.004 	|
> > > | 10 	| 0.019 	| 0.002 	|

---

### Official Review · Reviewer_cnAp · 2021-07-11

**Rating:** 7
**Confidence:** 3

**Summary:**

This paper tackles the problem of synthetic data generation for query release. First, it presents a general algorithmic framework (Adaptive measurements) for the task, and interprets a long line of prior work under this lense. At a high level, these works solve the min max problem in Equation 1.

Given this formulation, the paper proposes two algorithms, PEP and GEM. PEP selects a maximum entropy distribution to minimize the error for a given query set by solving the dual of the constrained optimization problem. They offer an iterative algorithm for this.  To avoid the exponential runtime of approaches like PEP (because they maintain a distribution over entire data domains), the paper also proposes GEM, which uses a generator network to represent this high dimensional space.  GEM can easily incorporate public data (with various missingness) by using it for pretraining.

Empirically, GEM outperforms prior work in accuracy to privacy tradeoffs on ACS and the adult datasets, and is more robust to domain shift in the public data than PMW or PEP. This gain is most pronounced when the data is high dimensional.


**Limitations And Societal Impact:**

The authors adequately note their limitations and possible ethical concerns (i.e known issues of DP and fairness).

**Main Review:**

Originality / Significance:
This paper unifies a large set of related algorithms under a single framework, and offers a useful new algorithm, GEM that improves over prior work in data generation for query release. GEM can naturally incorporate public datasets more robustly than prior work like PMWpub.  This study seems broadly valuable for data generation for query release.

Quality:
The experiments in this paper supported the core claims and adequately demonstrated the value of the method and when we would expect the method to help (i.e high dimensional data).

Clarity:
As a whole, the paper was fairly well written and has a detailed appendix, but I think it could benefit from expanded introductions through the main text  to make the work more accessible to the broader Neurips audience.

For instance, through the introduction, related work and abstract, the paper mentions MWEM without defining it, which makes the work less clear. This is only defined in page 4.  Giving a high level overview of how this method works earlier would make the paper more clear. Also, are there scenarios where leveraging PEP is preferred to GEM? It wasn’t clear to me through the results or discussion why both algorithms are highlighted, as GEM can both handle higher dimensional data and various public data sources.


**Time Spent Reviewing:**

2.5

---

> ### Author Response · Authors · 2021-08-10
> **Response to Reviewer cnAp**
>
> Thank you for your review and helpful suggestions.
>
> We will expand the introduction as you suggested to improve the clarity of our writing.
>
> Regarding your question about PEP, while both algorithms outperform existing benchmarks, there are cases in which PEP is the stronger algorithm, especially when the data dimension is low. In particular, PEP outperforms GEM on the ACS PA-18 (reduced) dataset. However, we concur that GEM is the more practical algorithm, and given its strong performance across a variety of settings, we would always recommend it.

---

### Official Review · Reviewer_Tcj3 · 2021-07-13

**Rating:** 8
**Confidence:** 3

**Summary:**

The authors unify several algorithms for DP synthetic data generation for query release. Using this framework they create two algorithms, private entropy projection (PEP) and generative networks with the exponential mechanism (GEM). Then they empirically evaluate their algorithms in publicly available datasets and show that their algorithms (mostly) outperform existing ones. They also show how GEM can use public data to improve its accuracy.


**Ethical Concerns:**

The authors mention that DP can cause unfair outcomes for certain groups, the Census Bureau example is discussed.

**Limitations And Societal Impact:**

Yes

**Main Review:**

Creating synthetic datasets for answering queries is an old and important problem in the DP literature. There exists excellent work on this problem and this paper not only manages to unify it but also improve upon it. I did not read all the proofs but what I read was technically correct. The paper very well written, it is easy to read and follow. Moreover, the relevant literature is discussed adequately. I also really appreciate the honest Limitations and Broader Impacts subsections.

Minor issues:
Line 91, typo. “Lete”
Line 149: “where describe” -> where we describe ?

**Time Spent Reviewing:**

3.5

---

> ### Author Response · Authors · 2021-08-10
> **Response to Reviewer Tcj3**
>
> Thank you for your positive and encouraging review. We will fix the typos mentioned.

---

### Official Review · Reviewer_aLmv · 2021-07-19

**Rating:** 7
**Confidence:** 3

**Summary:**

The paper has two main contributions:

1) It proposes a unifying framework called "Adaptive Measurements" that covers many differentially private synthetic data generation algorithms and show how these algorithms can be derived by modifying the loss function and distributional family.

2) Based on the above framework, two specific synthetic data generation algorithms called Private Entropy Projection (PEP) and Generative networks with Exponential Mechanism (GEM) have been proposed. Since PEP attempts to explicitly model the data distribution, it does not scale well to higher dimensions. GEM uses the well-known generative neural networks to overcome the above problem and can also effectively use public data for good initialization.

**Ethical Concerns:**

Not applicable.

**Limitations And Societal Impact:**

Yes, both the limitations and societal impact have been addressed adequately.

**Main Review:**

1) One of the main problems with the paper is that too many critical details have been pushed to the Appendix. For instance, the proof sketch of Theorem 1 must be included as part of the main text because it is not clear how the adaptive measurements mechanism ensures differential privacy. The paper should also explain what concentrated DP means from a practical standpoint and what level of privacy budget is acceptable. Similarly, the evaluation metric "max error" should also be defined and acceptable values of max error should be reported. Without such details, it would be very difficult for a practitioner to judge if the proposed algorithms are useful in real-world applications.

2) The differences between PEP and the MWEM schemes are not very clear. The paper simply brushes it off with the statement that adaptively assigning weights to past queries and measurements enables PEP to have faster convergence and better accuracy. What are the challenges and limitations in adaptive weight assignment?

3) What is the impact of the generator architecture G on the performance of the GEM scheme and how to choose the right architecture for a given problem?


**Time Spent Reviewing:**

3

---

> ### Author Response · Authors · 2021-08-10
> **Response to Reviewer aLmv**
>
> Thank you for your helpful comments. We address the comments in your main review below:
>
> 1) We will move the proof sketch of Theorem 1 to the main body.
>
>     We are mainly using zCDP as a tool for privacy loss accounting, since it enables a cleaner and tighter privacy analysis for compositions of the exponential mechanisms. All our main theoretical and experimental results are stated in terms of the standard (eps, delta) definition of DP.
>
>     We have defined max error in our preliminaries (see line 98). Regarding the question of what values are acceptable, we argue that this decision is up to the end-user, who must decide the trade-off between accuracy and privacy that they are willing to accept. Our goal is to provide practical algorithms that help users best optimize this trade-off.
>
> 2) A main difference between PEP and MWEM is that PEP naturally incorporates past queries and their corresponding measurements into its loss function by including each query measurement as a constraint in the constrained optimization problem. In addition, PEP adaptively (automatically) sets its learning rate such that the constraint for each query is satisfied, while MWEM uses a fixed learning rate that is relatively conservative in comparison to PEP. We will make this difference more clear in the main body of our paper when accounting for the improvement of PEP over MWEM.
>
> 3) Our goal was to show that even when using a simple multilayer perceptron architecture (that is the same across all experiments), we can achieve state of the art results for each dataset in the literature. It is possible that there exists a generator network that works better in practice for tabular data, but we leave further investigation to future work.

---

### Decision · Program_Chairs · 2021-09-27

**Decision:**

Accept (Poster)

**Comment:**

The paper got overall strong feedback from the reviewers. There were a few concerns regarding the presentation, once addressed will make the paper stronger. Some of those are:

1. A more formal treatment of the Private Entropy Projection algorithm. In particular the distinction between the constrained version, and the Lagrangian formulation.

2. The lack of theoretical analysis of utility. Since almost all the prior works have a formal utility analysis, it is important for the authors to point out any technical difficulty towards proving a formal analysis.